# Generalizable Active Learning: Boosting Out-of-Distribution Generalization in Active Learning with Simulated Generalization via Augmentation

## Abstract

Active Learning (AL) aims to select informative samples for annotation within a constrained labeling budget. Existing AL methods typically focus on achieving high performance on Independently and Identically Distributed (IID) source data and often terminate once IID performance converges. However, we identify a crucial yet under-explored issue: despite IID convergence, models trained under AL methods often exhibit a significant performance gap in Out-Of-Distribution (OOD) scenarios compared to models trained on the full labeled dataset, and closing this OOD gap often requires a much larger labeling budget. To address this issue, we introduce the task of **G**eneralizable **A**ctive **L**earning (GAL), which aims to improve the OOD generalization while preserving source-domain performance and minimizing additional labeling costs. We further introduce **Sim**ulated **G**eneralization **A**ctive **L**earning (SimGAL), a framework that simulates generalization scenarios through data augmentation without incurring extra annotations. SimGAL comprises: (1) **S**imulated **G**eneralization **A**ugmentation (SGA), which generates augmented samples simulating OOD characteristics for the pool of labeled samples, and (2) **Q**uality **S**tabilization **M**odule (QSM), which filters out overly distorted augmented samples to ensure stable training. We design two train-test paradigms specifically designed for the GAL task. Experimental results demonstrate that SimGAL significantly enhances OOD generalization performance of AL methods under matched labeling budget and training sample sizes.

## 1 Introduction

The remarkable success of deep neural networks is largely attributed to the availability of large-scale, high-quality labeled datasets (Sun et al., 2017). However, in real-world applications, acquiring sufficient labeled data remains a significant challenge. Most collected datasets are predominantly unlabeled, and the annotation process is typically expensive, time-consuming, and demands domain-specific expertise (Henaff, 2020). Under limited labeling budgets, efficiently leveraging unlabeled data has become a critical problem. Active Learning (AL) (Hsu & Lin, 2015) has emerged as a promising solution to this challenge. By incorporating sample selection into the training loop, AL iteratively selects and labels the most informative samples from a large pool of unlabeled data—typically

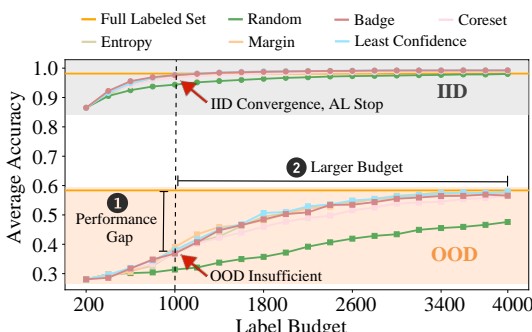

Figure 1: Performance curves of existing active learning methods and the fully labeled set on both IID and OOD domains under varying annotation budgets, evaluated on the Digits dataset.

guided by the source domain distribution—to progressively improve model performance (Feng et al., 2019). Compared to conventional fully supervised learning, AL significantly reduces annotation costs while achieving comparable performance on test domains that follow the Independent and Identically Distributed (IID) assumption (Settles, 2009b).

Most existing AL methods rely on uncertainty-based (Gal & Ghahramani, 2016; Zhu & Bento, 2017; Li et al., 2024) or diversity-based (Yoo & Kweon, 2019; Citovsky et al., 2021b; Li et al., 2023) strategies to select samples. These approaches primarily aim to improve performance on the IID source domain by fitting its underlying data distribution. However, in real-world applications, AL often operates under Out-Of-Distribution (OOD) scenarios—for instance, new weather and locations in autonomous driving or variations in scanners, protocols, and patient populations in medical imaging can introduce substantial distribution shifts. when AL evaluated on OOD domains—two notable phenomena emerge (as shown in Figure 1):

***Observation 1:*** When AL achieves performance convergence on IID data, its generalization capability under OOD scenarios often remains insufficient, exhibiting a noticeable performance gap compared to models trained on the entire fully labeled dataset. ***Observation 2:*** Compared to IID convergence, achieving OOD generalization convergence with AL requires a larger labeling budget.

These observations reveal a critical yet under-explored limitation: existing AL methods primarily prioritize performance on IID data and aim to maximize label efficiency (Zhang et al., 2020), but often inadvertently compromise the model's generalization ability under OOD scenarios. Bridging this OOD generalization gap typically demands a substantially larger labeling budget, which fundamentally counteracts the original motivation and cost-saving objective of AL.

To address this limitation, we introduce the **G**eneralizable **A**ctive **L**earning (GAL) task, which seeks to improve OOD generalization of AL methods while preserving source-domain performance under a limited labeling budget. Instead of acquiring new annotations, We explore an alternative strategy that embeds data augmentation into the AL loop to simulate OOD samples. This strategy aligns with the fundamental objective of AL, which is to minimize labeling cost. Guided by this intuition, we develop **Sim**ulated **G**eneralization **A**ctive **L**earning (SimGAL), a framework that integrates data augmentation into the AL pipeline. SimGAL consists of two key modules: **S**imulated **G**eneralization **A**ugmentation (SGA), which generates simulated OOD samples by dynamically combining standard data augmentation operations with learnable parameters; **Q**uality **S**tabilization **M**odule (QSM), which filters over-distorted samples based on semantic plausibility. To guide SGA, we introduce a novel loss function named SGA loss that encourages the augmented samples to deviate from their original class centers and exhibit greater classification uncertainty, while preserving semantic similarity. This guides the combination of augmentation operations to meaningfully approximate real OOD shifts, thereby generating plausible simulated OOD samples. Sample augmentation and parameter optimization are performed iteratively, where augmentation parameters are optimized by maximizing the SGA loss computed on the simulated OOD samples, and augmented samples are progressively updated until reaching the maximum number of iterations. To mitigate the potential negative effects caused by over-augmented samples, QSM identifies and filters semantically implausible augmented samples by approximating semantic plausibility through pixel statistics (mean and variance). Retained samples are added to the labeled pool, while filtered samples are re-augmented and re-evaluated until they meet QSM's quality criteria.

The main contributions can be summarized as follows:

- **Empirical Contribution.** We observed that although active learning achieved convergence in the IID domain, there is still a significant generalization gap on the OOD objective, and mitigating this gap requires a much larger label budget than that required for IID convergence.
- **Task Contribution.** We introduce the Generalizable Active Learning (GAL) task, which targets OOD generalization under limited labeling budgets while maintaining source-domain performance. To enable comprehensive evaluation, we construct two GAL train–test paradigms.
- **Technical Contribution.** We propose SimGAL, a novel framework that improves model robustness to distribution shifts by dynamically generating simulated OOD samples during active learning. To ensure stability, we design a quality stabilization module that detects and filters semantically implausible augmentations.

## 2 GENERALIZABLE ACTIVE LEARNING

In this section, we define the GAL task and analyze its challenges, in comparison with other active learning tasks. Due to space limitations, related work on active learning and a schematic illustration of the GAL task is provided in **Appendix A.1 and A.2**.

Table 1: Differences between GAL and other Active Learning tasks. A and B represent different domains or categories. B may also encompass multiple domains or categories.

| Task | AL stage | | Setting | | |
|---|---|---|---|---|---|
| | Unlabeled pool | Selected data | Training set | Test set | OOD shift |
| Active Transfer Learning | B | B | $A \cup B$ | B | ✓ |
| Multi-domain Active Learning | $A \cup B$ | $A \cup B$ | $A \cup B$ | $A \cup B$ | ✗ |
| Open-set Active Learning | $A \cup B$ | A | A | A | ✗ |
| Generalizable Active Learning | A | A | A | $A \cup B$ | ✓ |

## 2.1 ACTIVE LEARNING

In this work, we focus on the Active Learning (AL) paradigm, a widely adopted approach for data-efficient training. The setting consists of a large unlabeled data pool of size $N$, denoted as $U = \{\mathbf{x}_i\}_{i=1}^N$, where each instance $\mathbf{x}_i \in \mathcal{X}$ is a sample sampled from an underlying joint distribution $P(\mathcal{X}, \mathcal{Y})$. The objective is to learn a prediction model $f_\theta : \mathcal{X} \to \mathcal{Y}$ under a limited budget of $M$ labels, where $M \ll N$.

This process typically begins with a small set of randomly selected labeled samples, denoted as $L_0$. It then proceeds for $\mathcal{T}$ iterations. At each iteration $t$, the model $f_{\theta_t}$ is trained on the current labeled set $L_t$. A query strategy $\mathcal{Q}$ is then applied to select a batch of $b$ informative instances $B_t = \mathcal{Q}(U_t, f_{\theta_t})$ from the remaining unlabeled pool $U_t = U \setminus \{\mathbf{x} | \{\mathbf{x}, y\} \in L_t\}$. These selected instances are labeled by an oracle (*e.g.*, a human annotator) and added to the labeled set, $L_{t+1} = L_t \cup B_t$. The objective of AL is to design a query strategy $\mathcal{Q}$ that minimizes the expected risk on the IID test set, $D_{\text{test}}^{\text{IID}}$:

$$\underset{L_{\mathcal{T}} \subset U, |L_{\mathcal{T}}|=M}{\arg\min} \quad \mathbb{E}_{(\mathbf{x},y) \sim P(\mathcal{X},\mathcal{Y})} \big[ \ell(f_{\theta_{\mathcal{T}}}(\mathbf{x}), y) \big] \tag{1}$$

where $\ell(\cdot)$ is the loss function. The aim is to achieve performance comparable to a model trained on the fully labeled set, *i.e.*, $\text{ACC}(f_{\theta_{\mathcal{T}}}, D_{\text{test}}^{\text{IID}}) \approx \text{ACC}(f_{\theta_{\text{full}}}, D_{\text{test}}^{\text{IID}})$.

A fundamental limitation of the current AL paradigm lies in its inability to handle distributional shifts. Adapting robustly to significant distribution shifts in OOD scenarios remains a critical challenge.

## 2.2 GENERALIZABLE ACTIVE LEARNING

To address this limitation of AL, we introduce the Generalizable Active Learning (GAL) task. The GAL objective extends the AL paradigm by explicitly incorporating the demand for OOD generalization. The primary goal of GAL is to improve the OOD generalization while preserving source-domain performance and minimizing additional labeling costs. The procedural setup of GAL mirrors that of AL, involving iterative selection from an unlabeled pool $U$ under a budget $M^+$ (where $M^+ >= M$). However, the core optimization objective is redefined. The query strategy $\mathcal{Q}_{\text{GAL}}$ is designed not only to minimize the expected risk on IID test sets drawn from the same distribution $P(\mathcal{X}, \mathcal{Y})$ as the training set, but also to maximize performance on an OOD test set $D_{\text{test}}^{\text{OOD}}$, which is sampled from a different distribution $P_{\text{OOD}}(\mathcal{X}, \mathcal{Y}) \neq P(\mathcal{X}, \mathcal{Y})$. The ideal objective for GAL is thus:

$$\underset{L_{\mathcal{T}} \subset U, |L_{\mathcal{T}}|=M^+}{\arg\max} \quad \text{Performance}(f_{\theta_{\mathcal{T}}}, D_{\text{test}}^{\text{OOD}}) \tag{2}$$

In practice, we aim to select a labeled set $L_{\mathcal{T}}$ that satisfies two conditions simultaneously:

1. **IID Performance Maintenance**: $\text{ACC}(f_\theta, D_{\text{test}}^{\text{IID}}) \geq \tau$, where $\tau$ is a satisfactory performance threshold.

2. **OOD Performance Maximization**: Maximize the performance $\text{ACC}(f_\theta, D_{\text{test}}^{\text{OOD}})$.

The key challenge of GAL lies in enhancing the model's OOD generalization performance without incurring excessive additional labeling costs, while preserving its source-domain performance.

## 2.3 ANALYSIS OF GAL AND OTHER AL TASKS

The differences in GAL, Active Transfer Learning (ATL) (Wang et al., 2014), Multi-domain Active Learning (MDAL) (He et al., 2021), and Open-set Active Learning (OSAL) (Ning et al., 2022) tasks are shown in Table 1. ATL fine-tunes the model by selecting and labeling a small number of valuable samples from the target domain, aiming to enhance target-domain performance. However, it neither considers source-domain performance nor accounts for the fact that target domains are

Figure 2: The active learning cycle of the SimGAL framework. The red box highlights occurrences of extreme samples. $Max\mathcal{L}_{SGA}$ is designed based on the characteristics of real OOD samples, namely their high classification loss $\mathcal{L}_{cls}$ and large feature distance $d_c$. Furthermore, a constraint is introduced, where a smaller $d_s$ indicates a higher similarity.

often unavailable in real-world scenarios. MDAL focuses on sample selection across multiple source domains to enhance performance within those domains, while OSAL emphasizes identifying known classes from unlabeled data containing unknown categories to improve source-domain performance. Nevertheless, both MDAL and OSAL overlook performance under OOD conditions. In contrast, GAL not only maintains source-domain accuracy with limited annotations but also explicitly targets robustness to OOD data. This capability is especially critical in real-world scenarios.

## 3 METHODOLOGY

In this section, we present the motivation and implementation details of SimGAL framework, which is designed to address the challenge of GAL and improve OOD generalization without requiring additional annotations. The algorithmic flowchart is presented in **Appendix A.3.1**. The overall framework is shown in Figure 2, consisting of five key steps:

**Step 1: Data Selection.** At each AL round, we select $n$ samples from the unlabeled set $U_{\text{tr}}^t$. The first round is random; subsequent rounds is guided by the AL strategy using the model $f_\theta^{t-1}$. The selected set is denoted as $U_{\text{se}}^t = X^t$.

**Step 2: Annotation.** The selected unlabeled set $U_{\text{se}}^t$ in step 1 are annotated by oracle (*e.g.,* a human annotator) for labeling and are fed into the labeled set $L_{\text{tr}}^t$.

**Step 3: Simulated Generalization Augmentation (SGA).** We iteratively perform sample augmentation and augmentation parameter optimization. Specifically, we adopt random combinations of standard augmentation functions with learnable parameters, and guide the parameter optimization by maximizing the loss $\mathcal{L}_{SGA}$ until the maximum number of augmentation iterations is reached.

**Step 4: Quality Stabilization Module (QSM).** This step detects and filters extreme simulated OOD samples generated in Step 3. Samples that fail the detection are returned for re-augmentation and re-evaluation, and Steps 3 and 4 are repeated until all samples pass QSM's quality criteria and are forwarded to the subsequent steps.

**Step 5: Model Update.** The model $f_\theta^t$ is updated using the labeled set $L_{\text{tr}}^t$ together with the simulated OOD samples $L'_{\text{aug}}$. If $t < \mathcal{T}$, the process proceeds to the next round.

By applying both the SGA and QSM modules, we improve the OOD generalization robustness of models trained with limited labeling budget in OOD AL scenarios.

### 3.1 SIMULATED GENERALIZATION AUGMENTATION

The parameters of standard data augmentation functions play a critical role in image augmentation, particularly when multiple functions are applied in combination (Shorten & Khoshgoftaar, 2019). To enhance the diversity of simulated OOD samples, we construct an augmentation set consisting of standard data augmentation functions and randomly select a subset for combination in each augmentation. To better approximate real OOD samples, the parameters of these functions are made learnable and dynamically updated during iterative augmentation under the guidance of a loss function. This loss-based optimization adjusts perturbation intensity to produce more OOD-like samples, thereby improving the robustness of active learning models under limited labeling budget.

**Augmentation set combination.** We build an augmentation set consisting of several standard data augmentation operations, each controlled by learnable parameters. Each function controls a

specific type of distributional change in the sample. For example, a composite augmentation function can be written as $\psi(x; \omega) = o_r(o_c(x; \omega_c); \omega_r)$, where $o_c$ adjusts the contrast of $x$, $o_r$ modifies the scaling, and $\omega = \omega_c \cup \omega_r$ denotes the learnable parameters of $\psi$. We randomly generate $M$ augmentation compositions $\psi$, each drawn from the composition space $\Psi$. Each composition $\psi \in \Psi$ is formed by sequentially combining $L$ standard augmentation functions $\psi = \{o_i(\cdot; \omega_i)\}_{i=1}^{L}$, where $1 \leq L \leq L_{\max}$. Each composition induces a distinct distributional shift. $\psi$ is defined as:

$$\psi(x; \omega) = o_L(o_{L-1}(\cdots o_1(x; \omega_1) \cdots ; \omega_{L-1}); \omega_L) \tag{3}$$

where $o_i$ is a standard augmentation function parameterized by $\omega_i$. $\omega = \bigcup_{i=1}^{L} \omega_i$ is the union of the parameters of $L$ augmentation. Details of the augmentation set are provided in **Appendix A.3.2**.

Typically, longer augmentation combinations (larger $L$) can produce more diverse samples. However, excessively large $L$ may generate extreme samples that deviate significantly and incur higher computational costs. Since the target OOD domain is unknown, to strike a balance between augmenting diversity and reducing computational complexity, we fix a maximum augmentation length $L_{max}$ and uniformly sample the actual length $L$ of each augmentation combination from the discrete set $\{1, 2, \ldots, L_{max}\}$. The distribution space of augmentation combinations is $\Psi = ML_{max}$. $M$ satisfies the equation $M = \sum_{L=1}^{L_{\max}} M_L$, where $M_L$ denotes the number of combinations with length $L$.

**Learnable Augmentation Optimization**. The initial parameters of each standard augmentation operation are randomly set, leading to unstable augmentation effects. This makes it difficult to consistently and reliably generate simulated OOD samples, thereby limiting their potential to improve generalization. To address this, we make the augmentation parameters learnable. During the iterative augmentation process of SGA, we design a loss function that optimizes and updates the parameters by increasing the similarity between augmented samples and real OOD samples in the feature space, thereby making the generated samples more closely resemble the real OOD distribution. We provide an ablation study in **Appendix A.3.3** to analyze the effect of loss-based parameter optimization.

Samples from the OOD domain often exhibit significant discrepancies from those in the source domain, typically characterized by larger feature distances and higher classification losses (Yang et al., 2024). These two characteristics provide a clear direction for loss function design. To this end, we propose the SGA loss, which operates during the iterative process of composite augmentation. By incorporating a centrifugal mechanism, the SGA loss encourages the augmented samples to deviate from the class center, thereby increasing their feature distance. At the same time, it directly computes the classification loss and minimizes the distance to the original sample, ensuring semantic consistency. Ultimately, the SGA loss, designed according to real OOD distribution shifts, continuously optimizes the parameters of augmentation compositions, ensuring that the transformations meaningfully approximate actual OOD shifts.

To effectively apply the SGA loss during the iterative augmentation process, we design an augmentation strategy that progressively generates simulated OOD samples through learnable augmentation combinations. Since the augmentation process operates on batches, all samples within the same batch share the same augmentation combination. We denote the batch size as $b$. An excessively large $b$ may reduce the diversity of augmented samples, while an overly small $b$ may increase computational overhead. Therefore, we set $b$ to balance diversity and efficiency. The process can be described as follows. Given a labeled sample $\{x, y\}$, we construct an initial augmented sample $L'_m$ using a randomized set of augmentation operations and parameter ranges:

$$L'_m = \psi(x; \omega) \quad \omega = \bigcup_{j=1}^{L} \omega_j \sim \mathcal{U}(\omega_j^{\min}, \omega_j^{\max}) \tag{4}$$

where $(\omega_j^{\min}, \omega_j^{\max})$ is the initial parameter range for each augmentation operation.

We employ the model $f_\theta^{t-1}$, which has learned the parameter $\theta$ through active learning in the previous training round $t-1$, to compute the SGA loss of both the labeled sample $\{x, y\}$ and the initially augmented intermediate sample $L'_m$ in the current training cycle $t$. The optimization objective of the SGA loss is formulated as follows:

$$\mathcal{L}_{SGA} = \mathcal{L}_{cls}(L'_m, y) + \alpha d_c(L'_m, \bar{x}_y) - \lambda d_s(L'_m, x) \tag{5}$$

$$\min_{\theta \in \Theta} \max_{\omega \in \Omega} \mathbb{E}_{\psi \sim \Psi} \mathbb{E}_{x \sim L_{tr}} \mathcal{L}_{SGA} \tag{6}$$

where $\mathcal{L}_{cls}(L'_m, y)$ is the prediction loss for $L'_m$, $\Theta$ denotes the set of all possible values of $\theta$, $\Omega$ is the set of all possible values of $\omega$, $\alpha$ is a boolean parameter that flips its value in each augmentation iteration, *i.e.,* $\alpha =!\alpha$. $\lambda$ is a nonnegative regularization parameter. $d_c$ and $d_s$ is the squared euclidean distance function in the deep feature space of the model $f_\theta^{t-1}$, *i.e.,* $d_s(L'_m, x) = \|e' - e\|^2$ with the embeddings $e'$ and $e$ of $L'_m$ and $x$, respectively. $d_c(L'_m, \bar{x}_y) = |e' - \bar{e}_y|^2$, where $\bar{e}_y$ denotes the class-centered embedding of the labeled samples $L_{tr}^t$ for class $y$, which is recalculated from the newly acquired labeled samples after each AL round.

We adopt a continuous relaxation–based operation selection mechanism, which keeps the augmentation sampling process fully differentiable and enables efficient gradient-based updates of the combination parameters $\omega$, and repeat Eq. 4, 5, and 6 using the updated parameters. Loop through the maximum number of iterations $cyc$ to allow the parameters to learn sufficiently, thereby generating simulated OOD generalization samples that are as close as possible to real OOD samples.

The objective of Eq. 5 is to guide the direction of SGA by maximizing the SGA loss, such that the simulated OOD samples are pushed away from the class center in the feature space, thereby enhancing their category discrepancy from the original samples while preserving semantic consistency. Since the distribution of the training set varies across different active learning cycles, we do not explicitly constrain the distance when applying the SGA loss to encourage augmented samples to deviate from the training set center. Instead, we rely on the hyperparameter $cyc$, the similarity constraint $d_s$ in the loss, and the subsequent QSM mechanism to ensure the reliability of the simulated OOD samples.

## 3.2 QUALITY STABILIZATION MODULE

Data augmentation involves uncertainty (Wang et al., 2023). During the augmentation process, we found that optimizing augmentation parameters $\omega$ may cause deviation from the natural data distribution, resulting in perceptually abnormal extreme samples (such as overexposure, extreme darkness, saturation collapse, *etc.*). Such extreme samples not only lack semantic information but may also trigger gradient explosions or training oscillations. In **Appendix A.3.4**, we provide an analysis of the extreme samples generated by SGA and report that approximately **4–5%** of such cases emerge in each round of active learning iteration.

To mitigate the potential negative impacts from over-augmented samples, we propose the Quality Stabilization Module (QSM) as a crucial component of the SimGAL framework to maintain the stability of augmentation. The QSM filters out samples $L'_{aug}$ that deviate excessively from the simulated distribution, discards those that fail the detection, and re-simulates generalized augmentation for their original labeled samples until they pass the detection. All simulated OOD samples are provided for model training only after they have passed QSM detection.

We approximate the uncertainty and distributional similarity of simulated OOD samples using pixel-level statistical means and variances to support decision making. Accordingly, we define the detector $P_{ext} : x \to \{0, 1\}$ as follows:

$$P_{ext}(L'_{aug}) = \begin{cases} 1, & \text{if } \mu(L'_{aug}) \notin [I_{min}, I_{max}] \\ 1, & \text{if } \sigma(L'_{aug}) < V_{min}, \\ 0, & \text{otherwise.} \end{cases} \tag{7}$$

where $\mu(L'_{aug})$ and $\sigma(L'_{aug})$ denote the global pixel-level mean and variance of the symbol $L'_{aug}$, respectively. The intervals $[I_{min}, I_{max}]$ and the threshold $V_{min}$ represent perceptually reasonable bounds for brightness and contrast. Extreme pixel statistics, such as excessively variance or mean intensity outside reasonable bounds, often result in visually and semantically implausible samples; thus, monitoring these pixel-level metrics can effectively capture semantic consistency.

The QSM imposes a soft constraint on the composition space $\Psi$, limiting it to simulated OOD samples generated by perceptually valid compositions, *i.e.,* $\psi_{valid} \subset \Psi$:

$$\psi_{valid} = \left\{ \psi \in \Psi \mid \forall L'_{aug}, P_{ext}(L'_{aug}) = 0 \right\} \tag{8}$$

Re-augment and re-evaluate the generalization of samples $L'_{aug}$, ensuring $P_{ext}(L'_{aug}) = 0$, guaranteeing the validity of samples outside the simulated distribution.

Table 2: Comparison of OOD average generalization performance between SimGAL and AL under matched training samples and labeling budget settings. $N_L$ refers to the number of labeled samples, and $N_T$ refers to the number of training samples.

| Dataset | Number $N_L$ | $N_T$ | Method | Random | Margin | Leastconf | Entropy | Coreset | Badge | VAAL |
|---|---|---|---|---|---|---|---|---|---|---|
| Digits | 600 | 600 | Baseline | 30.19±1.59 | 30.60±2.06 | 31.95±1.75 | 30.98±1.28 | 31.63±1.49 | 31.74±0.83 | 31.50±0.93 |
| | 300 | 600 | +SimGAL | 52.99±1.96 | 52.26±1.67 | 54.35±2.15 | 52.63±2.28 | 51.09±1.66 | 53.22±3.50 | 53.09±1.54 |
| | 600 | 1200 | +SimGAL | **56.58±0.77** | **57.19±2.37** | **59.53±2.08** | **59.05±2.14** | **57.46±1.92** | **59.22±2.49** | **58.37±2.76** |
| | 800 | 800 | Baseline | 30.43±0.99 | 33.36±2.36 | 34.51±2.63 | 32.66±0.42 | 33.20±1.19 | 34.81±2.42 | 33.66±0.82 |
| | 400 | 800 | +SimGAL | 55.92±1.67 | 55.90±1.82 | 57.06±1.92 | 57.61±2.12 | 55.71±2.08 | 53.70±1.02 | 56.24±2.36 |
| | 800 | 1600 | +SimGAL | **61.25±1.24** | **60.74±1.92** | **62.41±0.85** | **61.39±1.81** | **60.05±0.67** | **60.09±2.08** | **61.77±1.96** |
| PACS | 400 | 400 | Baseline | 34.41±2.98 | 38.75±2.38 | 39.34±1.92 | 38.92±2.30 | 36.06±1.62 | 38.66±1.48 | 37.69±2.13 |
| | 200 | 400 | +SimGAL | 40.61±2.64 | 41.28±2.19 | 42.07±2.39 | 40.29±2.04 | 41.18±1.78 | 41.79±3.02 | 40.94±2.61 |
| | 400 | 800 | +SimGAL | **45.26±2.22** | **46.55±3.42** | **47.15±1.93** | **44.91±2.38** | **46.25±2.12** | **47.01±2.08** | **45.67±2.72** |
| | 600 | 600 | Baseline | 35.88±2.77 | 40.22±1.89 | 39.10±0.72 | 39.14±1.86 | 38.08±1.62 | 39.44±1.12 | 38.18±1.99 |
| | 300 | 600 | +SimGAL | 44.16±2.94 | 45.41±2.49 | 46.01±1.74 | 44.31±2.83 | 45.83±1.80 | 46.31±3.21 | 45.17±2.33 |
| | 600 | 1200 | +SimGAL | **50.37±3.00** | **49.55±1.63** | **51.16±2.59** | **49.37±2.15** | **50.74±1.92** | **51.96±2.10** | **50.29±3.02** |

## 4 EXPERIMENTS

### 4.1 GAL DATASET

To support the GAL task, we adopt two widely used datasets: Digits (Qiao et al., 2020) and PACS (Li et al., 2017). Digits is a composite digit recognition dataset comprising five domains: MNIST, SVHN, MNIST_M, SYN, and USPS. Each domain contains ten digit classes but varies in style, font, and color, reflecting distinct data distributions. Likewise, PACS consists of four distinct domains: Art Paintings, Cartoons, Photos, and Sketches. All domains share the same seven object categories but exhibit significant differences in visual style, leading to diverse distribution shifts. We design specific training-testing paradigms for both datasets. For Digits, MNIST serves as the IID source domain, while the remaining domains are designated as OOD test domains. For PACS, Photos serves as the IID source domain, and the other domains are used for OOD testing. Detailed descriptions of the datasets are provided in **Appendix A.4.1**.

### 4.2 EXPERIMENTAL SETTINGS

In this section, we present the main baselines and implementation details used in our study.

**Baselines.** We implement our proposed SimGAL framework based on several established AL baselines, including Random Sampling, Least Confidence (Leastconf) (Wang & Shang, 2014b), Margin (Balcan et al., 2007), Coreset (Sener & Savarese, 2017a), Entropy (Settles, 2009b), BADGE (Ash et al., 2019b), and VAAL (Sinha et al., 2019). Moreover, we instantiate SimGAL using Random Sampling as the query strategy and compare it with other active learning-based data augmentation methods, such as LADA (Kim et al., 2021) and CAMPAL (Yang et al., 2023). We further compare against classical data augmentation methods from the domain generalization (DG) literature, including ADA (Volpi et al., 2018), AutoAug (Cubuk et al., 2018), RandAug (Cubuk et al., 2020), and AdvST (Zheng et al., 2024). All methods are implemented under the same experimental settings.

**Implementation Details.** (1) **Digits experiment**: Since the Digits dataset consists of small-sized grayscale and color digit images, we follow the mainstream model practice of adopting lightweight networks (Zheng et al., 2024). we use LeNet (LeCun et al., 2002) as the backbone network. The initial labeling budget is set to 200, with 200 samples queried in each of the 20 active learning rounds. (2) **PACS experiment**: Due to the limited size and complexity of the PACS dataset, training a ResNet from scratch is prone to convergence issues. Therefore, we follow the mainstream model setting on this dataset (Chen et al., 2024). We use ResNet-18 (He et al., 2016) pretrained on ImageNet as the feature extractor backbone. The initial labeling budget is set to 200, and 200 samples are queried in each of the 5 active learning rounds. In all experiments, SimGAL simulates a number of samples equal to the labeled data in each cycle, effectively doubling the labeling data size. All experiments were performed using Nvidia RTX 2080 Ti GPUs. Each experiment was repeated four times with different random seeds, and we report the mean accuracy and standard deviation to ensure robustness. Best results are highlighted in bold. More details are provided in **Appendix A.4.2**.

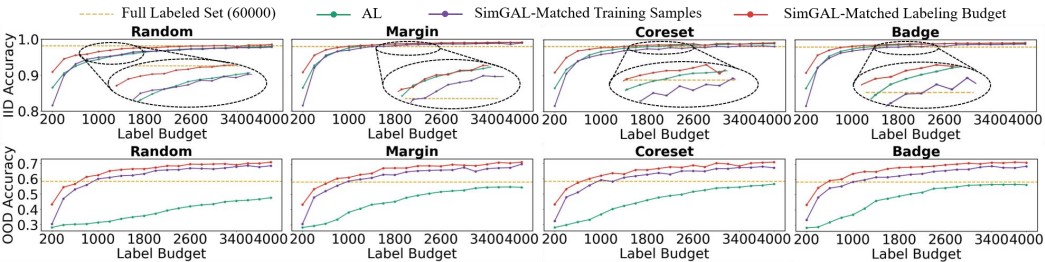

Figure 3: Comparison of IID source-domain and OOD generalization performance between SimGAL and AL on the Digits dataset under matched training samples and labeling budget. Please zoom in for a better view of the IID performance figure.

Table 3: Comparison of average generalization performance between SimGAL, active learning-based data augmentation methods, and domain generalization methods. All methods are implemented with random sample selection.

| Type | Method | Venue | Digits | | PACS | |
|---|---|---|---|---|---|---|
| | | | $N_L = 800$ | $N_L = 1000$ | $N_L = 800$ | $N_L = 1000$ |
| DG | AutoAug | CVPR'18 | 58.47±1.16 | 59.51±0.88 | 46.17±1.48 | 48.85±1.68 |
| | ADA | NeurIPS'18 | 33.29±1.67 | 35.22±1.86 | 36.32±1.15 | 37.61±1.96 |
| | RandAug | CVPR'20 | 60.22±0.58 | 61.32±1.03 | 47.34±1.06 | 49.71±1.42 |
| | AdvST | AAAI'24 | 57.53±1.97 | 60.43±0.79 | 50.92±2.03 | 52.43±1.32 |
| AL | LADA | NeurIPS'21 | 34.46±1.26 | 37.67±0.93 | 37.96±1.47 | 39.49±1.17 |
| | CAMPAL | ICML'23 | 45.78±2.18 | 49.27±1.76 | 42.18±1.79 | 44.86±1.45 |
| | SimGAL | - | **61.25±1.24** | **63.38±2.20** | **52.28±2.80** | **54.03±2.25** |

### 4.3 IMPROVEMENTS OF SIMGAL ON AL

In this section, we evaluate the impact of SimGAL on both IID performance and OOD performance under matched training sample sizes and labeling budget on the Digits and PACS datasets. As shown in Table 2 and Figure 3, the results demonstrate the remarkable performance of SimGAL in the GAL task. Results under various labeling budget ratios are provided in **Appendix A.4.3**.

**Comparison on the Digits Dataset.**

Under the same number of training samples $N_T$, SimGAL improves OOD performance by at least **19%**. Under the same labeling budget $N_L$, it achieves at least **25%** improvement in OOD generalization. Due to the inherent characteristics of AL, doubling the number of labeled samples does not necessarily lead to a proportional improvement in model performance. When the labeling budget is limited to $N_L = $ **600**, SimGAL applied with Entropy, BADGE, and Least Confidence sampling strategies, achieves OOD generalization accuracies of **59.05%**, **59.22%**, and **59.53%**, respectively—surpassing the performance achieved using the fully labeled set (**58.40%**) while utilizing only **1%** of the labeled samples. In contrast, standard AL methods require a labeling budget of $N_L = $ **4000** to achieve comparable OOD performance, demonstrating that SimGAL reduces the required labeling budget by about **85%**. In terms of source-domain performance, SimGAL not only maintains high accuracy under matched labeling budget, but also slightly improves the performance of AL methods. When the number of training samples is matched, SimGAL achieves comparable source-domain performance using only half labeling budget, with only a marginal performance drop.

**Comparison on the PACS Dataset.**

PACS is a challenging dataset characterized by substantial discrepancies across domains. Under the same number of training samples $N_T$, SimGAL improves OOD generalization performance by at least **2%**. Under the same labeling budget $N_L$, SimGAL achieves at least **6%** improvement in OOD generalization. As the labeling budget increases, SimGAL yields greater improvements in OOD generalization performance for AL methods. When the labeling budget is limited to $N_L = $ **200**, SimGAL applied with Random sampling achieves an OOD generalization accuracy of **40.61%**, surpassing the performance (**38.37%**) obtained using the fully labeled set. In contrast, active learning methods require labeling budget of $N_L = $ **400**, and in some cases up to $N_L = $ **600**, to reach similar OOD performance. This demonstrates that SimGAL reduces the required labeling budget by **50%**.

### 4.4 COMPARISON WITH AUGMENTATION-BASED AL AND DG METHODS

In this section, we compare our method's performance on OOD test domains against LADA and CAMPAL, which incorporate data augmentation into AL frameworks aiming to improve performance on the IID source domain. We also compare against domain generalization methods that rely on data augmentation strategies, including ADA, AdvST, RandAug, and AutoAug. As shown in Table 3, our method achieves the best performance on both the Digits and PACS datasets. Additional experimental results under varying data proportions are provided in **Appendix A.4.4**.

### 4.5 ABLATION STUDY

As shown in Table 4, we conduct an ablation study on the Digits dataset to evaluate the contributions of SGA and QSM. The baseline denotes AL with existing strategies evaluated on the OOD test domain. Adding SGA, which augments training with simulated OOD samples through learnable iterative combinations of augmentations, improves OOD generalization by over **21%**. Further integrating QSM, which filters extreme samples based on semantic plausibility, brings additional gains of **1.84%**, **2.93%**, and **5.25%** in different set-

Table 4: Ablation study on the Digits dataset. We report the average generalization performance across four OOD test domains when $N_L = 1000$.

| Module | | Margin | Entropy | Badge |
| SGA | QSM | | | |
| --- | --- | --- | --- | --- |
| × | × | 38.17±0.36 | 37.30±2.76 | 36.71±2.34 |
| ✓ | × | 59.48±1.50 ↑21.31 | 58.32±2.64 ↑21.02 | 60.87±1.63 ↑24.16 |
| ✓ | ✓ | **61.32±1.35** ↑1.84 | **63.57±1.74** ↑5.25 | **63.80±1.77** ↑2.93 |

tings. These results confirm that our method enhances AL's OOD generalization without extra labeling costs. Additional hyperparameter ablations are given in **Appendix A.4.5**.

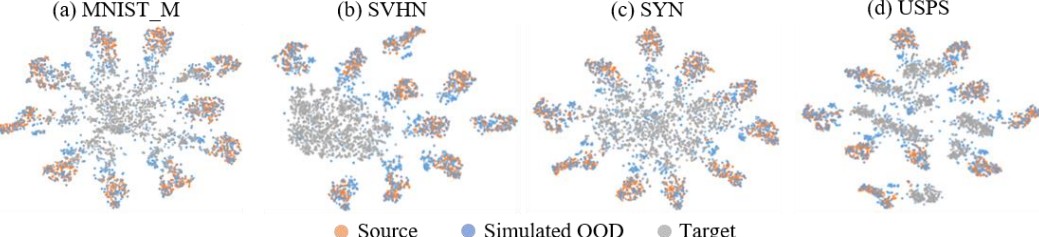

(a) MNIST_M  (b) SVHN  (c) SYN  (d) USPS

● Source  ● Simulated OOD  ● Target

Figure 4: Visualization of feature distributions for source domain, target domain, and simulated OOD samples. Four different target domain samples are randomly selected in equal quantity to match the source domain, and are visualized using t-SNE in the feature space under $N_L = 1200$.

### 4.6 VISUALIZATION

On the Digits dataset, we visualize the embedding space of source domain (orange, MNIST), simulated OOD samples (blue, by SimGAL), and four target domains (gray). As shown in Figure 4, the simulated OOD samples are distributed closer to the target domains while remaining near the source, effectively expanding its distribution without losing semantic integrity. A more detailed visualization comparing SimGAL and the Baseline is provided in the **Appendix A.4.6**.

## 5 CONCLUSION

In this paper, we identify a crucial yet under-explored issue: despite IID convergence, models trained under AL methods often exhibit a significant performance gap in OOD scenarios compared to models trained on the full labeled set, and achieving OOD convergence requires a substantially larger labeling budget. To address this issue, we propose the GAL task, which aims to improve the OOD generalization while preserving source-domain performance and minimizing additional labeling costs. To support this task, we design two dedicated training-testing paradigms for GAL. To address the GAL task, we propose the SimGAL framework, which seamlessly integrates data augmentation techniques that do not require additional labeled data into the AL process. It dynamically integrates standard augmentations with learnable parameters. A task-specific loss is designed to guide the generation of realistic OOD samples, while over-distorted ones are filtered based on uncertainty and distributional similarity. Our method improves the OOD generalization performance of AL without increasing the labeling budget, while preserving the performance on the source domain.

## REPRODUCIBILITY STATEMENT

First, our SimGAL framework clearly illustrates the entire active learning process and provides a detailed description. Second, in Section 3, we explicitly explain the construction of augmentation combinations, the formulations of SGA and QSM, and the iterative procedure of SGA. In the Appendix, we further provide the algorithmic diagram of SimGAL. Finally, in the experimental section, we describe the benchmark datasets and experimental setups in detail, and include the standard augmentation functions along with their parameter settings in the Appendix. With this comprehensive information, our work is fully reproducible.

## ETHICS STATEMENT

This work uses only publicly available datasets that do not contain sensitive personal information. The study does not involve human subjects, private data, or personally identifiable information. Our proposed method is developed for research purposes to advance generalizable active learning, and we do not foresee any direct negative societal or ethical impacts arising from this work.

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

# A    APPENDIX

## A.1    RELATED WORK

**Active Learning** focuses on selecting valuable samples for labeling from large datasets. Existing methods mainly use uncertainty-based or diversity-based criteria, or a combination of both (Ren et al., 2022).

Uncertainty-based active learning prioritizes samples about which the model is least confident, aiming to improve learning efficiency. Common strategies include measuring the margin between top class probabilities (Balcan et al., 2007), selecting samples with the lowest confidence scores (Wang & Shang, 2014a), and using entropy to quantify prediction uncertainty (Settles, 2009a). Advanced methods such as MC-Dropout (Gal & Ghahramani, 2016) and BALD (Gal et al., 2017) leverage Bayesian approximations to better estimate uncertainty, while approaches like BADGE use gradient information for sample selection (Ash et al., 2019a). Recent techniques also consider model output stability under noise as an uncertainty measure (Li et al., 2024).

Diversity-based methods focus on selecting samples that best represent the overall data distribution by considering differences in feature space. Common approaches include clustering (Citovsky et al., 2021a), where cluster centers guide sample selection, and VAAL (Sinha et al., 2019) learns the data distribution in the latent space using a VAE–GAN architecture and employs a discriminator to determine whether a sample belongs to the unlabeled distribution, thereby deriving an uncertainty-based sampling strategy. This reflects a typical Bayesian perspective: modeling the data distribution in the latent space to estimate the value of labeling. In contrast, our method does not aim to construct a Bayesian estimate of the data distribution. Instead, we focus on improving the robustness of AL under distribution shift: by using learnable augmentations to push samples toward the outer regions of their class manifolds and embedding these augmentations directly into the AL cycle, the two components reinforce each other, ultimately enhancing the AL mechanism itself. Graph Convolutional Networks have also been used to distinguish unlabeled samples differing from labeled ones (Caramalau et al., 2021). To reduce redundancy in iterative selection, distribution-based methods have been developed (Nguyen & Smeulders, 2004; Yang et al., 2015; Elhamifar et al., 2013; Hasan & Roy-Chowdhury, 2015). Notably, Coreset (Sener & Savarese, 2017b) selects subsets that effectively cover the entire data distribution.

Several approaches effectively combine uncertainty and representativeness for sample selection. For instance, CoreGCN (Caramalau et al., 2021) employs Graph Convolutional Networks to build a global data representation and integrates a core set strategy to select representative samples. Similarly, ClusterMargin (Citovsky et al., 2021b) applies hierarchical agglomerative clustering (HAC) to capture data diversity and then selects the most uncertain samples from different clusters. such as LALD (Kim et al., 2021) and CAMPAL (Yang et al., 2023). In the early stages of active learning, performance often suffers from instability due to limited labeled data, which can negatively affect subsequent sample selection. To address this, several works have integrated data augmentation into active learning. For example, LADA leverages data augmentation to enhance both classifier training and sample selection, thereby improving performance on the source domain. CAMPAL further explores how optimizing augmentation length can influence active learning effectiveness. While these methods improve performance within the source distribution, they often struggle to generalize effectively to Out-Of-Distribution (OOD) scenarios.

In recent years, (Deng et al., 2023) observed fluctuations between IID and OOD performance in AL under NLP settings, which mainly stem from the discrete nature of text manifolds, unstable semantic boundaries, and high sampling noise. These factors make active sampling more prone to inducing training distribution drift, resulting in pronounced oscillations across AL rounds. Moreover, since the IID performance in their setting never truly converges throughout the AL process, they were unable to clearly isolate the structural OOD gap that persists even after IID convergence. Besides, several studies (Margatina et al., 2021; Yuan et al., 2022; Chen et al., 2024) have applied active learning to multi-domain generalization tasks. These studies mainly focus on the impact of distribution differences between similar samples in different domains on generalization performance. This fundamentally differs from our work on GAL tasks, which simulates OOD samples from a single source domain to improve the generalization performance of active learning in the presence of OOD domains.

**Data augmentation** is widely used to improve model generalization. Traditional methods (*e.g.,* (Hendrycks et al., 2019b; Zoph et al., 2020)) enhance source-domain performance through random or hybrid transformations, but struggle to generate samples with large domain shifts. Mixup (Zhang et al., 2017) improves generalization by interpolating samples within the source domain. Adversarial augmentations ((Volpi et al., 2018; Zhao et al., 2020)) attempt to create challenging examples in pixel or feature space, but still lack diversity in domain shifts. Methods like advST (Zheng et al., 2024) use learnable transformations to simulate worst-case domain shifts. GANs and VAEs (Qiao et al., 2020; Wang et al., 2021) have also been explored, but generated samples often remain in-distribution due to training limitations.

Although our augmentations do not originate from real OOD domains, they push samples away from the original source distribution along semantically preserved directions, thereby inducing distributional shifts. Such distribution-shift–based augmentations have been widely regarded in prior works(Tobin et al., 2017; Hendrycks et al., 2019a; Choi et al., 2023) as an effective form of synthetic OOD data, and have been systematically adopted and validated in studies on OOD generalization, domain randomization, and robustness.

## A.2 GAL VS. AL

In this section, we introduce the essential differences between GAL tasks and AL. As shown in Figure 5, the overall performance of AL on both IID and OOD data resembles a mountain-climbing trajectory. As the labeling budget increases, the overall performance steadily improves, reaching the peak at a budget of 4000. Conventional AL methods typically focus on fitting the source domain, achieving convergence on IID performance around a budget of 1000—effectively halting halfway up the mountain. In doing so, they largely ignore the model's generalization ability to OOD data, which is critical for real-world deployment. In contrast, the GAL task emphasizes convergence on OOD generalization performance, ultimately leading to the summit.

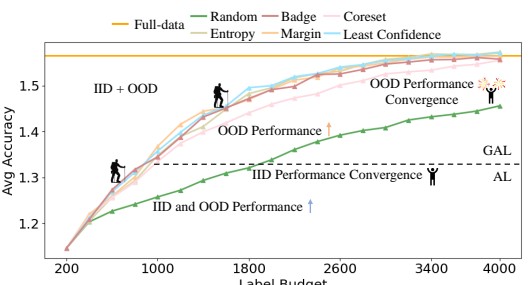

Figure 5: Comprehensive performance curves of AL method under different budgets on OOD and IID

In the early stages of active learning, the inherent limitation of having only a small number of labeled samples—combined with a persistent tendency to select difficult samples—often leads to a bias phenomenon. Intuitively, fitting a model with a limited labeled set already introduces bias, and the heuristic strategies in active learning that preferentially select hard examples further amplify the distributional discrepancy between the training data and the overall data population, making this discrepancy more extreme and effectively introducing statistical bias (Chen et al., 2025). As a result, active learning tends to overfit the source-domain distribution, leading to insufficient generalization when evaluated on test domains that exhibit distributional shifts from the source.

## A.3 METHOD SUPPLEMENT

### A.3.1 SIMGAL ALGORITHM DIAGRAM

Algorithm 1 illustrates the overall workflow of SimGAL. The process begins with a randomly selected initial labeled set $L_{\text{tr}}^0$ from the unlabeled training pool $U_{\text{tr}}$, which is used to train the initial model $f_w^0$. In each active learning round $t$, the query strategy $\mathcal{Q}$ selects $B_{\text{query}}$ samples from the current unlabeled set $U_{\text{tr}}^{t-1}$ for annotation. These newly labeled samples form the updated labeled set $L_{\text{tr}}^t$. SGA process then follows. Using a random combination of parameterized augmentations $\psi(x; \omega)$, SGA generates candidate augmented samples. These are iteratively optimized via the augmentation combination parameters over $cyc$ steps, where the objective $\mathcal{L}_{aug}$ encourages augmented samples to deviate from their original class centers while preserving semantic consistency. The learned parameters $\omega$ aim to produce effective simulated OOD samples that challenge the model. After optimization, the final augmented set $L'_{\text{aug}}$ is validated through the extrapolation predictor $P_{ext}$. If the criterion is not met,

---

**Algorithm 1** SimGAL Framework

---

**Require:** Unlabeled training set $U_{\text{tr}}$, labeled training set $L_{\text{tr}}$ , a query strategy $\mathcal{Q}$, initial budget $B_{\text{init}}$, query budget $B_{\text{query}}$, iteration round $\mathcal{T}$, number of augmentation iterations $cyc$, a standard augmentation $o(x;\omega)$, intermediate samples $L'_{\text{m}}$ during iterative augmentation simulated OOD samples $L'_{\text{aug}}$, decision function $P_{ext}$ for QSM, the model $f_w^t$ trained in the $t$-th round.
1: Randomly select $B_{\text{init}}$ samples from $U_{\text{tr}}$, add them to $L_{\text{tr}}^0$, and train the initial model $f_w^0$ on $L_{\text{tr}}^0$:
2:  $U_{\text{tr}}^0 \leftarrow U_{\text{tr}} \setminus \{X | \{X, Y\} \in L_{\text{tr}}^0\}$
3: **for** $t = 1, 2, ..., \mathcal{T}$ **do**
4:   Select $B_{\text{query}}$ samples and annotate $Y^t$:
5:    $Y^t \leftarrow \text{Annotate}(\mathcal{Q}(U_{\text{tr}}^{t-1}, B_{\text{query}}))$
6:    $L_{\text{tr}}^t \leftarrow L_{\text{tr}}^{t-1} + \{X^t, Y^t\}$
7:    $U_{\text{tr}}^t \leftarrow U_{\text{tr}} \setminus \{X | \{X, Y\} \in L_{\text{tr}}^t\}$
8:   Random augmentation combination:
9:    $\psi(x; \omega^0) \leftarrow o_L(o_{L-1}(\cdots o_1(x; \omega_1) \cdots ; \omega_{L-1}); \omega_L)$ in Eq.3
10:  **for** $c = 1, 2, ..., cyc$ **do**
11:    A augmentation operation:
12:     $L'_{\text{m}} \leftarrow \psi(L_{\text{tr}}^t; \omega^{c-1})$ in Eq.4
13:    Compute SGA loss:
14:     $d_c(L'_m, \bar{x}_y) \leftarrow \|L'_m - \bar{x}_y\|^2$
15:     $\mathcal{L}_{aug} \leftarrow \mathcal{L}(L'_m, y) + \alpha d_c(L'_m, \bar{x}_y) - \lambda d_s(L'_m, x)$ in Eq.5
16:    Optimization objective:
17:     $\min_{\theta \in \Theta} \max_{\omega \in \Omega} \mathbb{E}_{\psi \sim G} \mathbb{E}_{x \sim L_{tr}} \mathcal{L}_{aug}$ in Eq.6
18:    Parameter update:
19:     $\omega^c = \text{Update}(\omega^{c-1})$
20:  **end for**
21:  $L'_{\text{aug}} \leftarrow \psi(L_{\text{tr}}^t, \omega^{cyc})$
22:  **if** $P_{ext}(L'_{\text{aug}}) = 1$ in Eq.7 **then**
23:    **go to Step 8**
24:  **end if**
25:  Model update:
26:  $f_w^t \leftarrow \text{Update}(f_w^{t-1}, L_{\text{tr}}^t + L'_{\text{aug}})$
27: **end for**
28: **return** $f_w^{\mathcal{T}}$

---

the augmentation process restarts. Otherwise, the model $f_w^t$ is updated using both the labeled and augmented data. This loop repeats for $\mathcal{T}$ rounds, resulting in a robust, domain-aware model $f_w^{\mathcal{T}}$.

### A.3.2 AUGMENTATION SET IN SGA

As shown in Table 5, we adopt a set of 11 standard image augmentation operations in the SimGAL framework. Each operation is parameterized by a variable $w$, sampled from a predefined uniform distribution $\mathcal{U}(a, b)$. These augmentations include common image transformations such as rotation, translation, scaling, and contrast adjustment, which are essential for improving the model's generalization capability under limited labeled data. Additionally, some operations like solarization, posterization, and histogram equalization are included to simulate more diverse visual patterns, promoting robustness to semantic-preserving perturbations. The set covers both photometric and geometric transformations, offering a comprehensive and flexible augmentation space for SimGAL.

Table 6: Average generalization performance on digits using random parameters and learnable parameters.

| Method | $N_L = 800$ | $N_L = 1000$ |
|---|---|---|
| Random | 57.00±0.40 | 59.90±1.61 |
| Learned(%) | **61.25±1.24** | **63.38±2.02** |

### A.3.3 ANALYSIS OF THE SGA

The SGA is embedded in the AL, and the two are not independent of each other, but rather mutually reinforcing. First, in the early stages when AL data is extremely scarce, the model's estimation of unlabeled samples is often not robust enough. Embedding augmentation into the AL process

Table 5: The standard Augmentation operation table used in this experiments. The letter $x$ denotes given images. The $w$ denotes the parameter of the augmentation operation. $\mathcal{U}(a, b)$ denotes a continuous uniform distribution at interval [a, b].

| Augmentation | Parameters | Description |
|---|---|---|
| HSV($x$,$w$) | $w \sim \mathcal{U}(-1, 1)$ | Adjusts hue, saturation, and brightness |
| Rotate($x$,$w$) | $w \sim \mathcal{U}(0.01, 1.0)$ | Adjusts the rotation magnitude |
| Translate($x$,$w$) | $w \sim \mathcal{U}(-1, 1)$ | Controls the translation range along the x, y axes |
| Invert($x$,$w$) | $w \sim \mathcal{U}(0.5, 1.0)$ | Inverts the image |
| Shear($x$,$w$) | $w \sim \mathcal{U}(-0.3, 0.3)$ | Shear angle along the x and y axis |
| Contrast($x$,$w$) | $w \sim \mathcal{U}(0.1, 1.9)$ | Adjusts image contrast |
| Sharpness($x$,$w$) | $w \sim \mathcal{U}(0, 1.0)$ | Sharpens the image |
| Solarize($x$,$w$) | $w \sim \mathcal{U}(-0.5, 0.5)$ | Solarizing the image |
| Scale($x$,$w$) | $w \sim \mathcal{U}(0.5, 2.0)$ | Scales the image |
| Equalize($x$,$w$) | | Equalize the image histogram |
| Posterize($x$,$w$) | $w \sim \mathcal{U}(\{0, 1..., 8\})$ | Posterizing the image |

Table 7: Comparison of average cross-domain generalization performance on the Digits dataset between SimGAL and the "AL-first-then-augmentation" strategy.

| Method | $N_L = 400$ | $N_L = 600$ | $N_L = 800$ | $N_L = 1000$ |
|---|---|---|---|---|
| AL-first-then-augmentation | 50.07±2.15 | 55.38±1.65 | 58.20±0.95 | 59.41±1.21 |
| SimGAL | **53.56±3.02** | **57.19±2.37** | **60.74±1.92** | **61.32±1.35** |

can stabilize the model's estimation before each round of sample selection, fundamentally avoiding the cumulative bias caused by early incorrect estimations in subsequent rounds. Second, the samples selected by AL usually have the highest information value. Applying augmentation to these key samples can amplify their contribution, generating samples with higher generalization gain, thereby further improving the selection quality of AL. Therefore, our method utilizes a dynamic coupling mechanism of "selection-augmentation-reselection," rather than a simple serial process. Table 7 shows that the DG-type augmentation can interact with AL. Performing data augmentation after completing the AL process does lead to performance improvements, but the gains remain clearly inferior to those achieved when augmentation is integrated into the AL loop.

We present the motivation behind introducing learnable parameters into the SGA module. We compare the impact of augmentation combinations on OOD generalization performance under both random and learnable settings. As shown in Table 6, incorporating learnable parameters enables more effective utilization of augmentation operations, thereby achieving further performance improvements. Simply put, without learnable parameters, the reinforcement process becomes unstable, and the parameters cannot be effectively leveraged.

Table 8: The proportion of extreme samples generated by SGA and detected by QSM in different cycles relative to the labeled samples in that cycle.

| Round | t=1 | t=2 | t=3 | t=4 | t=5 |
|---|---|---|---|---|---|
| Number(%) | 5.00 | 4.75 | 5.00 | 4.25 | 4.30 |

### A.3.4 THE MOTIVATION OF QSM

As shown in Figure 6, we observe that the simulated OOD samples generated by SGA contain semantically implausible extreme cases. An analysis of these samples reveals that, at the pixel statistical level, both their mean and variance exhibit similar characteristics. Since the semantically implausible extreme samples exhibit distinctive characteristics in pixel statistics, we design the QSM module to detect and filter

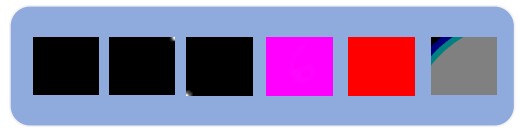

Figure 6: Illustrative examples of some extreme augmented samples.

out such harmful examples. As shown in Table 8, we report the percentage of extreme samples detected by QSM across five active learning rounds, relative to the number of labeled samples in each round. The ratio consistently stays around 4%–5%, indicating that QSM can effectively and stably identify a small portion of detrimental samples. This mechanism helps mitigate their negative impact and supports better generalization during training.

## A.4 ADDITIONAL EXPERIMENTS

### A.4.1 DATASET DETAILS

Table 9: Comparison of average generalization performance between SimGAL, active learning-based data augmentation methods, and domain generalization methods. All methods are implemented with random sample selection.

| Type | Method | Venue | Digits | | PACS | |
|------|--------|-------|--------|--------|--------|--------|
| | | | $N_L = 400$ | $N_L = 600$ | $N_L = 400$ | $N_L = 600$ |
| DG | AutoAug | CVPR'18 | 54.04± 0.59 | 56.24±0.40 | 40.95±1.36 | 45.27±0.84 |
| | ADA | NeurIPS'18 | 30.88±1.25 | 31.45±1.25 | 35.17±1.30 | 36.03±0.79 |
| | RandAug | CVPR'20 | 52.28±1.56 | **56.71±1.09** | 42.04±1.22 | 46.92±2.31 |
| | AdvST | AAAI'24 | 48.96±2.11 | 55.05±1.61 | 44.63±1.68 | 49.19±1.51 |
| AL | LADA | NeurIPS'21 | 36.25±1.46 | 37.24±1.69 | 36.25±1.46 | 37.24±1.69 |
| | CAMPAL | ICML'23 | 39.37±1.38 | 42.67±0.83 | 39.01±2.01 | 40.46±0.94 |
| | SimGAL | - | **54.48±3.33** | 56.58±0.77 | **45.26±2.22** | **50.37±3.00** |

The **Digits** dataset used in our experiments is a composite digit recognition benchmark comprising five distinct domains: MNIST, SVHN, MNIST_M, SYN, and USPS. All five domains share the same label space (digits 0–9), but differ significantly in image style, resolution, background complexity, and color distribution. Specifically, MNIST consists of 60,000 grayscale handwritten digits for training and 10,000 for testing; SVHN contains over 73,000 colored digit images; MNIST_M includes 59,001 training and 9,001 testing samples blended with natural background patches; SYN provides 479,400 synthetic digit images with varied styles; USPS contains 7,291 training and 2,007 testing grayscale digits originally in $16 \times 16$ resolution. For consistency, we resize all digit images to $32 \times 32$ and normalize them to the $[0, 1]$ range. The **PACS** dataset is a domain generalization benchmark in object recognition, composed of four stylistically distinct domains: Art Paintings, Cartoons, Photos, and Sketches. Each domain shares the same seven object classes: *dog, elephant, giraffe, guitar, horse, house, and person*. The sample sizes vary: Art contains 2,048 images, Cartoon 2,344, Photo 1,677, and Sketch 3,929 images. All images are resized to $224 \times 224$ and normalized using ImageNet statistics to match the input requirements of ResNet-18.

### A.4.2 IMPLEMENTATION DETAILS

**Digits Experiment.**

In this section we describe in detail the parameters used in our experiments. For the Digits dataset, we adopt a consistent training configuration across all compared methods to ensure fairness. All images are resized to 32×32 pixels and converted to three-channel RGB format. The training is conducted for 50 epochs with a batch size of 64. The maximum number of augmentation learning cycles in the SGA module is set to cyc = 20, and the regularization coefficient is $\lambda = 10$. The learning rate is initialized at $lr = 0.0001$, and reduced by a factor of 0.1 after 25 epochs to encourage convergence. These hyperparameters are empirically selected

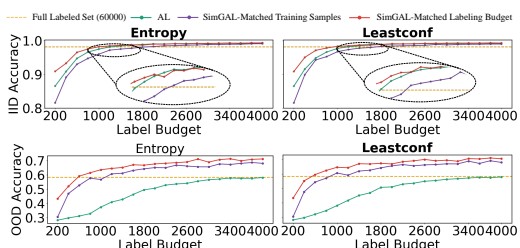

Figure 7: Comparison of IID source-domain and OOD generalization performance between Sim-GAL and AL on the Digits dataset under matched training samples and labeling budget. Please zoom in for a better view of the IID performance figure.

to balance model stability and generalization under limited labeling budgets. **PACS Experiment.** In

Table 11: Comparison of the average generalization performance between SimGAL and semi-supervised methods on the Digits dataset.

| Method | $N_L = 400$ | $N_L = 600$ | $N_L = 800$ | $N_L = 1000$ |
|---|---|---|---|---|
| StyleMatch | 42.18±1.42 | 45.72±1.74 | 47.61±2.32 | 49.26±0.94 |
| SimGAL | **53.56±3.02** | **57.19±2.37** | **60.74±1.92** | **61.32±1.35** |

the PACS experiment, we employ a ResNet-18 backbone pre-trained on ImageNet to leverage transfer learning. The model is trained for 50 epochs with a batch size of 32. We set the maximum augmentation learning iterations to $\text{aug}_{max} = 20$, and the regularization coefficient to $\lambda = 10$, consistent with the Digits setup. The learning rate is initialized at $lr = 0.001$ and adjusted dynamically using a cosine annealing scheduler, enabling smooth convergence and reducing overfitting in later training stages. The Digits and PACS datasets we chose are widely used benchmark datasets in generalization research. Digits uses the lightweight network LeNet to avoid overfitting due to an excessively large network. Due to the limited size and lower visual complexity of the PACS dataset, we use ResNet-18 pre-trained on the currently mainstream ImageNet dataset. The pretrained backbone primarily supplies generic low-level and mid-level visual features, ensuring that the model starts from a strong and well-structured initial representation. The source domain selection follows the settings of existing works on these two datasets. Regarding the label budget setting, we use the label budget required for AL convergence on each dataset. On Digits, a budget of 1000 achieves ID convergence, so we use a setting of 5 cycles with an interval of 200. Since PACS has a smaller dataset, we maintain the same settings as Digits. When aligning the labels, we use the same label budget as the benchmark, but the total sample size is twice that of the benchmark. The aligned sample size is half that of the benchmark, but the total sample size is the same as the benchmark. This is to maintain relative fairness with the benchmark at the same label budget and absolute fairness with the same total sample size.

### A.4.3 OTHER AL EXPERIMENTS

As shown in Figure 7, we further evaluate the performance of SimGAL when applied to various mainstream AL methods. The results demonstrate that SimGAL consistently improves model performance across different AL strategies. This observation highlights the general applicability and robustness of SimGAL in improving OOD generalization under diverse AL settings. Regardless of the underlying AL algorithm, SimGAL delivers consistent performance gains, verifying its effectiveness in broader application scenarios. As shown in Table 11, our method outperforms the semi-supervised approaches. Although using unlabeled data increases the number of training samples via augmentation, these samples share the same distribution as the labeled data and therefore cannot effectively improve OOD generalization.

| Length | 1 | 2 | 3 | 4 |
|---|---|---|---|---|
| SimGAL | 58.42±1.11 | 59.99±3.22 | 62.40±1.24 | **63.38±2.02** |
| time(h) | 0.4 | 0.5 | 0.6 | 0.8 |

| Length | 5 | 6 | 7 | 8 |
|---|---|---|---|---|
| SimGAL | 62.98±0.62 | 61.78±1.88 | 63.06±1.33 | 62.01±1.80 |
| time(h) | 1.1 | 1.2 | 1.3 | 1.6 |

Table 10: The average generalization performance of SimGAL on the Digits dataset with $N_L = 1000$ under different augmentation combination lengths $L$ and the runtime for four repetitions over five rounds. The results are split into two sub-tables for clarity.

### A.4.4 EXPERIMENTS WITH OTHER RATIOS

In this section, we compare our method with LADA and CAMPAL under various domain ratio settings. Both LADA and CAMPAL incorporate data augmentation into active learning frameworks to enhance performance on IID source domains. We also evaluate our method against several domain generalization techniques—ADA, AdvST, RandAug, and AutoAug—which aim to improve model robustness and generalization by introducing diverse or challenging augmentations through adversarial or randomized strategies. As shown in Table 9, our method consistently outperforms all baselines across different domain proportion settings on both the Digits and PACS datasets.

### A.4.5 ABLATION EXPERIMENTS ON HYPERPARAMETERS

In this section, we analyze the influence of the hyperparameters employed in SimGAL. Different augmentation combination lengths lead to varying degrees of image transformation, and the interactions among augmentation operations also differ. As shown in Table 10, we analyze the impact of the maximum standard augmentation combination length on the performance of SimGAL. The results indicate that a length of 4 yields the best performance. When the combination length is too short, the diversity of augmentations is limited, resulting in insufficient distribution shift of the simulated OOD samples. In contrast, an overly long combination increases the risk of over-augmentation, which can negatively affect the model's performance. As shown in Table 12, we further analyze the impact of the maximum augmentation iteration count in the SGA module on the performance of SimGAL. The experimental results demonstrate that the best performance is achieved when the maximum iteration count is set to 20, indicating that the augmentation learning process is sufficiently thorough. When the iteration count is too low, the parameters of the augmentation combinations are undertrained and fail to effectively approximate OOD samples. Conversely, if the iteration count is too high, the parameters may become overfitted or drift, leading to a slight decrease in performance.

Table 12: Ablation experiment of the maximum enhancement iteration cycle of SGA. The results are split into two parts for clarity.

| cyc | 5 | 10 | 15 | 20 | 25 | 30 | 35 |
|---|---|---|---|---|---|---|---|
| SimGAL | 60.83±1.80 | 61.8±2.00 | 62.74±1.91 | **63.38±2.02** | 62.77±2.19 | 61.86±1.15 | 61.57±1.34 |

In addition, as shown in the table 13, 14, we examine the impact of the parameters $\alpha$ and $\lambda$ in the SGA loss on model generalization. The parameter $\alpha$ controls the dynamic repulsion that pushes augmented samples away from the class center. Without $\alpha$, augmented samples cannot be effectively driven out of their local neighborhood, weakening SGA's ability to expand

Table 13: Ablation study on parameter $\alpha$ in the SGA loss

| $\alpha$ | 0 | 0 or 1 | 1 |
|---|---|---|---|
| SimGAL | 60.83±1.92 | **63.38±2.02** | 62.14±1.48 |

semantic coverage. Conversely, using a fixed $\alpha$ forces continuous repulsion, which may distort or drift the semantic content of the samples. The parameter $\lambda$ balances augmentation strength and semantic preservation. An excessively large $\lambda$ enforces overly strong similarity constraints between augmented samples and the original samples, which hinders the distributional expansion of the augmented data. Conversely, when $\lambda$ is too small, the similarity constraint becomes weak, making it difficult to preserve the semantic content of the samples.

Table 14: Ablation study on parameter $\lambda$ in the SGA loss.

| $\lambda$ | 7 | 8 | 9 | 10 | 11 | 12 | 13 |
|---|---|---|---|---|---|---|---|
| SimGAL | 61.13±2.80 | 62.11±1.32 | 62.87±2.01 | **63.38±2.02** | 62.59±1.39 | 61.71±1.42 | 60.19±1.67 |

### A.4.6 VISUALIZATION

Table 15: The average center feature distance between SimGAL and AL for 10 categories in 5 domains on the Digits dataset. A smaller value indicates a higher concentration of samples in the same domain.

| Methon | 200 | 400 | 600 | 800 | 1000 |
|---|---|---|---|---|---|
| Baseline | 22.92 | 34.60 | 35.28 | 36.28 | 37.47 |
| +SimGAL | **15.89** | **20.91** | **33.06** | **28.49** | **35.94** |

To evaluate the impact of SimGAL on AL, we conducted a comparative visualization study on the Digits dataset. We visualized and compared the feature distributions of the same class across different domains in the embedding space using the SimGAL framework and a standard AL method. As shown in Figure 8, we selected samples from three classes across five domains and plotted their

t-SNE distributions. The results show that under the AL method, feature distributions across domains are more dispersed, whereas our method yields more compact and centralized distributions. To further validate this observation, we quantified the inter-domain feature center distances for different classes. As reported in Table 15, our method consistently achieves smaller inter-domain distances across all AL cycles. A smaller distance indicates tighter intra-class clustering, which correlates with better generalization performance. These visualization results demonstrate that our method effectively expands the diversity of sample distributions and enhances the model's sensitivity to domain variations, thereby improving its robustness when faced with OOD samples.

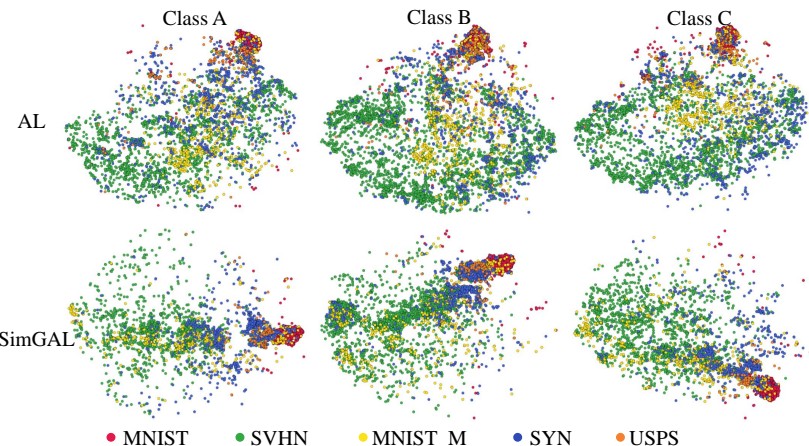

Figure 8: t-SNE visualization of feature distributions for three classes across five domains in the Digits dataset, comparing models trained by SimGAL and AL with $N_L = 1000$ labeled samples.

## LLM DISCLAIMER

In this paper, Large Language Models (LLM) were used solely after the completion of the entire manuscript, and only for proofreading and polishing to improve its fluency and readability. All observations, ideas, experiments, as well as the writing of the manuscript and the preparation of figures, were entirely conducted and completed by the authors without the assistance of LLM.

