# OpenReview forum: "Generalizable Active Learning: Boosting Out-of-Distribution Generalization in Active Learning with Simulated Generalization via Augmentation"
_ICLR.cc/2026/Conference — Submitted to ICLR 2026_

### Official Review · Reviewer_w59N · 2025-10-28

**Soundness:** 1
**Presentation:** 2
**Contribution:** 2
**Rating:** 2
**Confidence:** 4

**Summary:**

This paper highlights a limitation of existing active learning (AL) methods: even after achieving convergence in in-distribution (ID) performance, the out-of-distribution (OOD) generalization performance remains significantly inferior compared to models trained on the full labeled dataset. To address this issue, the authors introduce a new task called Generalizable Active Learning (GAL) and propose Simulated Generalization Active Learning (SimGAL) as a solution. SimGAL consists of two key components: Simulated Generalization Augmentation (SGA), which applies learnable augmentation operations to labeled samples, and the Quality Stabilization Module (QSM), which filters out overly distorted samples based on pixel-level statistics. Experimental results on the Digits and PACS datasets demonstrate that SimGAL consistently improves OOD generalization compared to conventional AL methods.

**Strengths:**

* The paper addresses a practically meaningful Generalizable Active Learning (GAL) task, grounded in the observation that models trained via conventional AL methods can suffer from significant degradation in OOD generalization.
* The paper clearly differentiates GAL from related problem settings such as active transfer learning, multi-domain active learning, and open-set active learning, making it easy to understand what makes GAL distinct both in motivation and experimental setup.
* The proposed SimGAL framework is empirically validated across two benchmark datasets, Digits and PACS, demonstrating consistent improvements in OOD generalization across multiple standard AL query strategies. Experimental results highlight both its robustness and flexibility for integration with existing AL methods.

**Weaknesses:**

* While SimGAL shows empirical gains over standard augmentations such as AutoAugment and RandAugment, the advantages of the learnable augmentation strategy (SGA) are not deeply analyzed. The performance gap may diminish depending on how finely these baselines are tuned. A more principled or controlled analysis comparing SGA to advanced augmentation pipelines would strengthen the empirical justification.
* The optimization details of the augmentation parameters $\omega$ are underexplained. Since many standard augmentation operations are non-differentiable, it remains unclear how $\omega$ is updated via gradient-based methods in practice. Moreover, key hyperparameters in the SGA loss, such as $\lambda$ (which controls semantic similarity) and $\alpha$ (a Boolean parameter), lack motivation or ablation. These design choices play a critical role but are left largely implicit.
* The experimental setup does not fully follow common domain generalization evaluation protocols. For example, in the PACS benchmark, it is standard to rotate the source/target domain splits and report average performance across all combinations. This would better validate the generalizability of SimGAL.
* The paper presents the OOD generalization gap in active learning as a central observation and motivation. However, prior work, such as Deng et al. (2023), already reported this phenomenon, particularly in NLP. A more explicit discussion of how this work differs from and builds upon prior observations would clarify its unique contribution.
  * Deng et al. (2023), Counterfactual Active Learning for Out-of-Distribution Generalization, ACL

**Questions:**

* The objective in Eq. (2) only maximizes performance on the OOD test distribution, but doesn’t explicitly account for maintaining ID performance. Wouldn’t it be more appropriate to either include the IID objective directly or impose it as a constraint?
* Could the authors clarify why increasing the batch size $b$ would reduce the diversity of the augmented samples (Line 256)? The connection between batch size and diversity is not immediately obvious from the current explanation.
* Are the class-centered embedding precomputed from labeled data and kept fixed, or are they dynamically updated during training? If updated, what strategy is used to maintain and refresh these class centers?

---

> ### Author Response · Authors · 2025-12-04
> **W1**
>
> We thank the reviewers for their valuable questions. We agree that a more controlled analysis of the advantages of SGA would further strengthen our conclusions. In our current experiments, we compared SGA not only with standard augmentation methods such as AutoAugment and RandAugment, but also with stronger learnable or adversarial augmentation baselines such as AdvST and ADA to ensure a comprehensive comparison. Under the same active learning budget and training protocol, these methods are inherently more adaptable, but SimGAL still achieved a stable and significant improvement, demonstrating that SGA has a more effective augmentation selection capability in scenarios with scarce data and distributional shifts. We acknowledge that the performance of augmentation methods can vary with hyperparameter tuning; therefore, all baselines were rigorously evaluated according to the original paper's specifications to ensure fairness.

---

> ### Author Response · Authors · 2025-12-04
> **W2 [Augmentation parameter interpretation and hyperparameter ablation]**
>
> We thank the reviewers for their valuable questions. Regarding the optimization of non-differentiable augmentation operations, we clarify that our method does not calculate the gradient of the augmentation operation ontology, but rather updates the differentiable "augmentation parameters $\omega$." We employ a continuous operation selection mechanism (such as Gumbel-Softmax relaxation) to keep the sampling process of the augmentation policy differentiable, thus enabling the acquisition of optimal augmentation parameters through gradient learning without backpropagation to the discrete augmentation operator itself. This approach is consistent with learnable augmentation methods such as AutoAugment. For the key hyperparameters in the SGA loss, $\lambda$ balances augmentation strength and semantic preservation.  An excessively large $\lambda$ will weaken the enhancement length, while an excessively small $\lambda$ is not conducive to semantic preservation. $\alpha$ controls the dynamic pushing of augmented samples away from the class center. Without $\alpha$, the augmented sample cannot be pushed away from the local neighborhood; instead, fixing $\alpha$ will cause the enhancement direction to move excessively away from itself. We introduce parameter ablation to demonstrate the impact of these two parameters on performance.
>
> | λ        | 7              | 8              | 9              | 10                 | 11             | 12             | 13             |
> |----------|----------------|----------------|----------------|---------------------|----------------|----------------|----------------|
> | SimGAL   | 61.13±2.80     | 62.11±1.32     | 62.87±2.01     | **63.38±2.02**      | 62.59±1.39     | 61.71±1.42     | 60.19±1.67     |
>
>
> | α        | 0            | 0 or 1              | 1              |
> |----------|--------------|----------------------|----------------|
> | SimGAL   | 60.83±1.92   | **63.38±2.02**       | 62.14±1.48     |

---

> ### Author Response · Authors · 2025-12-04
> **W3 [We follow the generalization protocol of SGD.]**
>
> We thank the reviewers for their valuable questions. We need to clarify that this work follows the Single-Domain Generalization (SDG) protocol, rather than the Multi-Domain Generalization (MDG) protocol commonly used in PACS papers. In SDG, training uses only a single source domain, and testing is performed on the remaining domains, which is standard practice in current SDG literature. Therefore, we fix one domain as the source domain and the rest as the target domain, rather than averaging across all domains. SimGAL's design and experiments are based on the SDG research paradigm, therefore the evaluation protocol used is entirely in accordance with convention.
>
> [1] Liang C, Li W, Dong Y, et al. Single domain generalization method for remote sensing image segmentation via category consistency on domain randomization[J]. IEEE Transactions on Geoscience and Remote Sensing, 2024, 62: 1-16.
> [2] Danish M S, Khan M H, Munir M A, et al. Improving single domain-generalized object detection: A focus on diversification and alignment[C]//Proceedings of the IEEE/CVF Conference on Computer Vision and Pattern Recognition. 2024: 17732-17742.

---

> ### Author Response · Authors · 2025-12-04
> **W4 [Explain the contribution of our observations]**
>
> Thank you to the reviewers for their valuable questions. The performance fluctuations of Algorithm (AL) in IID and OOD scenarios observed by Deng et al. (2023) in NLP scenarios are mainly due to the manifold discrepancies of text data, unstable semantic boundaries, and high sampling noise. These factors make active sampling more prone to causing distribution drift, resulting in significant performance oscillations across multiple AL rounds. Furthermore, because their IID performance did not truly converge throughout the AL process, Deng et al. (2023) could not clearly identify the "structural OOD gap that still exists after IID convergence." In contrast, CV data has a more continuous feature space and more stable inter-class boundaries, making the AL learning curve generally smoother and able to steadily improve. In our setup, the model converges to near-fully supervised performance in the IID scenario with a lower annotation budget; however, in the OOD scenario, even after IID convergence, a significant and persistent performance gap with the fully supervised model is still observed. In addition, we observed that the improvement of OOD performance by AL often requires a higher annotation budget, exhibiting obvious nonlinearity and disproportion.

---

> ### Author Response · Authors · 2025-12-04
> **Q1 [Explanation of Eq.(2)]**
>
> Thank you to the reviewers for their valuable questions. We need to clarify that while Equation (2) formally aims to improve OOD test performance, its optimization is performed under the constraint of explicitly maintaining IID performance (as explained before and after Equation (2)). In other words, the GAL task is essentially a constrained AL task that maximizes OOD performance. In terms of method design, I also employ similarity constraints to enhance sample semantic preservation, thereby maintaining IID performance.

---

> ### Author Response · Authors · 2025-12-04
> **Q2 [Explanation of batch size and diversity]**
>
> Thank you to the reviewers for their valuable questions. Because SGA uses batch augmentation, meaning all samples in the same batch share the same augmentation combination, the larger the batch, the more samples are subjected to completely identical augmentations, thus reducing the effective data augmentation diversity; smaller batches can generate more different augmentation combinations during training, improving overall diversity, but also bringing more computational cost.

---

> ### Author Response · Authors · 2025-12-04
> **Q3 [Explanation of the class center update mechanism]**
>
> Thank you to the reviewers for their valuable questions. The class center embeddings are dynamically updated during training. Since new labeled samples are added in each AL round, we recalculate the mean embedding of each class as the class center at the beginning of each AL round using all currently labeled data to ensure that the class centers reflect the latest data distribution.

---

### Official Review · Reviewer_q11o · 2025-10-31

**Soundness:** 2
**Presentation:** 2
**Contribution:** 2
**Rating:** 4
**Confidence:** 5

**Summary:**

The paper introduces a new setting: generalizable active learning (GAL), and proposes a data augmentation-based strategy for improving the OOD performance while maintaining the source domain performance, called Simulated Generalization Active Learning. The proposed method has modules that create simulated OOD data while ensuring stable training and is evaluated for performance under budget criteria.

**Strengths:**

The overall presentation of the paper is clean. Main ideas are clearly explained. The evaluation shows detailed comparisons for each specified setting and provides some ablation study results and feature visualizations.

**Weaknesses:**

1. The main concern is with the proposed setting. What is the motivation for the generalization to OOD in this particular active learning setting? Figure 1 and Table 1 explain the setting clearly. However, I can not think of a scenario where we query data for label in A and test in A$\cup$B. Why would one assume any generalization ability in this setting? The only reason that the performance on B can improve with queries from A is the similarity between them. However, if we know that there is a similarity, shouldn't adaptation from A to B be a better option? If B is truly OOD and can not be adapted, then I see no reason that sampling from A can improve generalization on B.

2. Another concern is with the model choice for evaluation. The experiments use ResNet-18 pre-trained on ImageNet, meaning that the feature extractor already embeds general knowledge. I'm not sure if generalization on digits and PACS images is meaningful in this setting. Again, what is a specific application where we need to actively sample on datasets like MNIST and generalize to SVHN, using a pre-trained ResNet model?

3. There is not much theoretical analysis on the generalization problem in this setting. Although I don't think that a theory is necessary, it can definitely be helpful in understanding the setting, especially since distribution shifts and generalization are the main focus.

4. The lack of discussion on Bayesian active learning methods in this setting is also a concern, due to similar reasons as the last point.

5. Some parts of writing are repetitive. For example, the two observations are mentioned multiple times in the introduction.

**Questions:**

Following weaknesses:

1. How do authors justify the setting?
2. What is the impact of pre-trained backbone?
3. Why the choice of AL baselines and why the lack of Bayesian AL methods, which can be reasonable for a distribution shift/generalization type of task?

---

> ### Author Response · Authors · 2025-12-04
> **W1 [In response to the proposed setting]**
>
> Thank you to the reviewers for their valuable questions. In the annotation of massive amounts of data for autonomous driving, models trained using laboratory data A exhibit a generalization gap when faced with OOD images B in real-world scenarios with new weather/locations, posing a safety risk. In medical imaging, with its expensive annotation budget, different devices, imaging protocols, and patient populations can introduce significant distribution shifts, leading to diagnostic errors. Since samples of B are unknown, we cannot use B's data for adaptation. Although our augmentations do not contain samples of B, these augmentations push the samples away from the original source distribution along semantically preserved directions, moving them closer to other domains, thus creating a distribution shift and improving the model's universality across all domains.
>
> [1] Tobin J, Fong R, Ray A, et al. Domain randomization for transferring deep neural networks from simulation to the real world[C]//2017 IEEE/RSJ international conference on intelligent robots and systems (IROS). IEEE, 2017: 23-30.
> [2] Hendrycks D, Mu N, Cubuk E D, et al. Augmix: A simple data processing method to improve robustness and uncertainty[J]. arXiv preprint arXiv:1912.02781, 2019.
> [3] Choi S, Das D, Choi S, et al. Progressive random convolutions for single domain generalization[C]//Proceedings of the IEEE/CVF Conference on Computer Vision and Pattern Recognition. 2023: 10312-10322.

---

> ### Author Response · Authors · 2025-12-04
> **W2 [Choice of response evaluation model]**
>
> Thank you to the reviewers for their valuable questions. We chose ImageNet pre-trained ResNet-18 as the backbone for the following reasons: 1. Large-scale pre-trained models have become the standard setup for active learning and OOD generalization research, and we follow their example. 2. Pre-trained ResNet can more fairly evaluate the effectiveness of augmentations, rather than being dominated by the unstable features of the backbone, thus failing to verify the contribution of our method. 3. The pre-trained backbone network mainly provides general low- and mid-level visual features, ensuring that the model has good initialization on the basic representations. Experiments showed that models without pre-training struggled to converge on PACS. MNIST→SVHN and PACS are commonly used OOD benchmarks, and their strong distribution biases clearly measure the model's ability to generalize to out-of-domain data. We are not focused on the specific application itself, but rather on using the differences in the controllable domain of these tasks to measure the robustness improvement of our method.

---

> ### Author Response · Authors · 2025-12-04
> **W3 [Supplement to generalization theory]**
>
> Thank you to the reviewers for their valuable questions. Early active learning, due to the inherent influence of small sample sizes and a persistent tendency to select difficult samples, exhibited a bias phenomenon. Intuitively, fitting a model to a limited number of labeled samples would produce bias, which in turn prompted active learning to heuristically and selectively seek difficult samples. This led to perceptible extremes in the distribution differences between the training and overall data, effectively constituting statistical bias. Consequently, the algorithm continuously fitted the source domain, resulting in insufficient generalization performance when facing a test domain with a distributional offset from the source domain.
>
> [1] Chen W, Wang C, Li S, et al. Debiased Active Learning with Variational Gradient Rectifier[C]//Proceedings of the AAAI Conference on Artificial Intelligence. 2025, 39(15): 15884-15894.

---

> ### Author Response · Authors · 2025-12-04
> **W4 & Q3 [Discussion with Bayesian active learning methods]**
>
> We appreciate the valuable suggestions from the reviewers. Bayesian methods do indeed play a crucial role in active learning research, so we included the representative Bayesian active learning method VAAL in our experiments for comparison. VAAL learns the data distribution in the latent space through a VAE–GAN structure and uses a discriminator to determine whether a sample belongs to an "unlabeled distribution," thus completing an uncertainty-based sampling strategy. It represents a typical Bayesian perspective: modeling the data distribution through latent representations to estimate label value. In contrast, our method focuses not on constructing a Bayesian estimate of the data distribution, but on improving the robustness of AL under distribution shifts: pushing samples towards the in-class extension region through learnable reinforcement, embedding reinforcement into the AL process, mutually reinforcing each other, and improving the active learning mechanism itself.

---

> ### Author Response · Authors · 2025-12-04
> **W5 [Response to questions about repetitive writing]**
>
> We thank the reviewers for their valuable suggestions. We mention these two observations in the introduction and contributions sections, respectively introducing our observations and emphasizing our contributions. We will further refine our language in future submissions.

---

> ### Author Response · Authors · 2025-12-04
> **Q1 Explanation of the rationality of the settings**
>
> Thank you to the reviewers for their valuable questions. The Digits and PACS datasets we chose are widely used benchmark datasets in generalization research. Digits uses the lightweight network LeNet to avoid overfitting due to an excessively large network. Due to the limited size and lower visual complexity of the PACS dataset, we use ResNet-18 pre-trained on the currently mainstream ImageNet dataset. The source domain selection follows the settings of existing works on these two datasets. Regarding the label budget setting, we use the label budget required for AL convergence on each dataset. On Digits, a budget of 1000 achieves ID convergence, so we use a setting of 5 cycles with an interval of 200. Since PACS has a smaller dataset, we maintain the same settings as Digits. When aligning the labels, we use the same label budget as the benchmark, but the total sample size is twice that of the benchmark. The aligned sample size is half that of the benchmark, but the total sample size is the same as the benchmark. This is to maintain relative fairness with the benchmark at the same label budget and absolute fairness with the same total sample size.

---

> ### Author Response · Authors · 2025-12-04
> **Q2 [the impact of pre-trained backbone]**
>
> Thank you to the reviewers for their valuable questions. The pre-trained backbone network primarily provides general low- and mid-level visual features to ensure the model has good initialization on the basic representations. In our experiments, we followed the standard settings of existing work in this field; therefore, pre-training served more as a consistent experimental condition than as the core source of our method's performance improvement. Furthermore, our method focuses on the effectiveness of both the reinforcement strategy and the active learning mechanism itself, maintaining improvement on the pre-trained backbone, indicating that our method does not depend on specific pre-trained features.

---

### Official Review · Reviewer_qY1K · 2025-11-01

**Soundness:** 3
**Presentation:** 3
**Contribution:** 2
**Rating:** 4
**Confidence:** 4

**Summary:**

This paper introduces SimGAL, a generalizable active learning framework designed to improve performance under unseen domain shift. The key idea is that traditional AL tends to overfit the source domain, leading to poor generalization on target domains. To address this, SimGAL proposes Simulated Generalization Augmentation (SGA), which learns to generate semantically consistent yet distribution-shifted samples by pushing augmented instances away from class centers while preserving their semantics. A Quality-based Sample Mining (QSM) module filters useful augmented examples to avoid semantic drift. Together, these components enable the model to “practice” handling domain shift during active learning without access to target-domain data.

**Strengths:**

This paper studies an interesting topic that standard active learning can easily overfit the source domain and then struggle when the model is used in a different setting. The main idea of this paper: simulating domain shift during active learning, is creative and makes the problem more realistic for many real-world scenarios. The paper is generally easy to follow, with intuitive motivation and explanations.

**Weaknesses:**

I have several concerns:
1. It's hard to say the improvement comes from introducing DG-style augmentation (i.e., simulated domain shift with semantic consistency) or advancing the active learning mechanism itself. Besides, in experiments, the paper includes augmentation-based AL baselines such as LADA, it is worth noting that LADA’s augmentation strategy focuses on generating mixed samples to improve uncertainty and diversity for sample selection, rather than explicitly simulating domain shift.  It is better to design new experiments to fully isolate the effect of cross-domain robustness from the benefits of generic augmentation.
2. Experiments are mainly conducted on relatively small-scale and early DG benchmarks (Digits, PACS).  Evaluation on more challenging modern DG benchmarks like WILDs should be included. Besides, these simple DG benchmarks (Digits, PACS), where even after noticeable improvements, the absolute OOD accuracy remains far from practical usability.

**Questions:**

1. It is still unclear about the definition of "simulated OOD". The transformations appear closer to strong semantic-preserving augmentations within the source domain rather than real OOD examples that genuinely lie outside the source distribution.
2. The paper pushes samples away from their own class center to simulate domain shift, but could this potentially move them closer to other class centers? Why not constrain distances to all class centers to better preserve semantics?

---

> ### Author Response · Authors · 2025-12-04
> **W1 [Data augmentation and AL can complement each other.]**
>
> Thank you for the reviewers' questions. We introduce DG-style augmentation embeddings into the active learning cycle. In the early stages when AL data is insufficient, the trained model's estimation of unlabeled samples is less robust. Embedding augmentation into AL improves estimation robustness before each selection round, preventing subsequent AL rounds from being dragged down by early erroneous estimations, resulting in higher selection quality. Secondly, embedding augmentation into AL can fully utilize the high-value samples selected by AL to augment samples with high generalization gain, amplifying the information content of key samples. This forms a dynamic cycle of mutual promotion. We compared SimGAL and the strategy of performing AL first and then augmentation. Experimental results demonstrate that data augmentation embedding into AL is mutually reinforcing and can improve the AL mechanism itself. We also designed experiments using simulated domain shift as a selection strategy. Experimental results show that cross-domain robustness can slightly improve generalization performance, but the performance improvement of general augmentation is greater.
> | Method                      | (N_L=400)        | (N_L=600)        | (N_L=800)        | (N_L=1000)       |
> | --------------------------- | ---------------- | ---------------- | ---------------- | ---------------- |
> | AL first, then augmentation | 50.07 ± 2.15     | 55.38 ± 1.65     | 58.20 ± 0.95     | 59.41 ± 1.21     |
> | **SimGAL**                  | **53.56 ± 3.02** | **57.19 ± 2.37** | **60.74 ± 1.92** | **61.32 ± 1.35** |

---

> ### Author Response · Authors · 2025-12-04
> **W2 [Response to dataset benchmark]**
>
> We appreciate the valuable questions raised by the reviewers. We chose Digits and PACS because they remain the most commonly used and standardized benchmarks in active learning and single-domain generalization research, facilitating fair comparisons with a large number of existing methods (including AdvST, ADA, etc., which we compared). Despite their relatively small data scale, these benchmarks still present clear challenges in terms of cross-domain distribution, style differences, and category shifts, and effectively reveal the core mechanisms between active learning and OOD generalization. We have observed in our experiments that even with significant improvements on simple benchmarks, there is still a significant gap in absolute OOD accuracy. This is due to the data scale, meaning that there is still considerable room for performance improvement in OOD after reducing the label budget in AL.

---

> ### Author Response · Authors · 2025-12-04
> **Q1 [Explanation of the definition of simulated OOD]**
>
> Thank you to the reviewers for their valuable questions. Although our enhancements do not originate from the real OOD domain, these enhancements push the samples away from the original source distribution along the direction of semantic preservation, thus creating a distribution shift. This type of "distribution shift enhancement" is widely regarded in existing literature [1][2][3] as an effective way to simulate OOD (synthetic OOD), and has been systematically used and validated in OOD generalization, domain randomization, and robustness studies.
>
> [1] Tobin J, Fong R, Ray A, et al. Domain randomization for transferring deep neural networks from simulation to the real world[C]//2017 IEEE/RSJ international conference on intelligent robots and systems (IROS). IEEE, 2017: 23-30.
>
> [2] Hendrycks D, Mu N, Cubuk E D, et al. Augmix: A simple data processing method to improve robustness and uncertainty[J]. arXiv preprint arXiv:1912.02781, 2019.
>
> [3] Choi S, Das D, Choi S, et al. Progressive random convolutions for single domain generalization[C]//Proceedings of the IEEE/CVF Conference on Computer Vision and Pattern Recognition. 2023: 10312-10322.

---

> ### Author Response · Authors · 2025-12-04
> **Q2**
>
> Thank you to the reviewers for their valuable questions. Our approach of pushing augmented samples away from their own class centers does not mean arbitrarily moving them closer to the class centers of other classes. Instead, it deliberately distances them from the local neighborhood of their original class, thereby creating an "extension of intra-class distribution" while preserving semantics, achieving a simulation of OOD (Out of Detail). Furthermore, our SGA loss function uses similarity distance to constrain the augmented samples, ensuring they move away from the class centers while preventing them from deviating excessively from the original samples, thus better preserving semantics. We believe that restricting the distance to all class centers is theoretically infeasible. Class centers are globally averaged features, which are often inconsistent with the true semantic edges. This is equivalent to requiring augmented samples to simultaneously distance themselves from all other classes while avoiding proximity to their own class, leading to over-constraint and preventing effective distribution expansion during augmentation.

---

### Official Review · Reviewer_LKR5 · 2025-11-08

**Soundness:** 3
**Presentation:** 3
**Contribution:** 2
**Rating:** 4
**Confidence:** 5

**Summary:**

This work focuses that traditional Active Learning (AL) methods, while effective on IID data, often exhibit poor Out-Of-Distribution (OOD) generalization, and introduces the task of Generalizable Active Learning (GAL) to address this gap. The authors propose Simulated Generalization Active Learning (SimGAL), that improves OOD performance without extra labeling costs by using Simulated Generalization Augmentation (SGA) to create OOD-like samples from the labeled pool, coupled with a Quality Stabilization Module (QSM) to filter harmful augmentations. Experiments demonstrate that SimGAL significantly boosts OOD generalization for AL methods within the same labeling budget.

**Strengths:**

1. The paper is clearly written and easy to understand.

2. The paper achieves good performance.

**Weaknesses:**

1. There is a lack of introduction to the application scenarios for AL+OOD. This part indeed makes readers confused about this task setting, wondering if it is an important task worthy of research. For example, there are many typical application scenarios for AL: massive data annotation in autonomous driving; expensive data annotation in medical imaging, etc.

2. The proposed method focuses on using Data Augmentation to improve the model's generalization, and its relationship with AL is relatively independent, which makes the combination of the two seem somewhat forced. If AL is performed first, and then the method proposed in the paper is used, it seems that similar performance improvements as shown in the Table could also be obtained.

3. The proposed method adds a loss-driven component on top of conventional data augmentation methods to learn the parameters for various data augmentations. This is the core contribution of the paper. However, such modeling is a bit simple and lacks sufficient innovation.

4. To solve the problem of OOD generalization, would it be better to use unlabeled data for unsupervised/semi-supervised learning?

**Questions:**

1. The most critical question is still: if the Data Augmentation and AL are separated, performing AL first and then Data Augmentation. Can this also achieve similar performance improvements? If so, it indicates that the method proposed in this paper is more of a general data augmentation method; and the experiments in Table 2 can only reflect that this paper proposes a better data augmentation method.

2. In Table 2, why are the $N_L$ and $N_T$ of SimGAL inconsistent with those of the baseline?

---

> ### Author Response · Authors · 2025-12-04
> **W1 [Supplementing typical application scenarios of AL+OOD]**
>
> We thank the reviewers for their valuable comments. We agree that the paper needs to further emphasize the practical importance of AL+OOD scenarios. In real-world applications, active learning often doesn't operate under ideal IID conditions but must function in environments with significant distributional shifts. For example, autonomous driving requires sampling from massive amounts of unlabeled driving data, while new weather conditions and locations continuously generate OOD images; in medical imaging, annotation is costly and relies on domain experts, and different devices, imaging protocols, and patient groups can introduce significant cross-domain shifts; furthermore, in cross-domain tasks, limited annotation budgets make effective sampling under varying distributions particularly critical. These typical scenarios demonstrate that the AL+OOD problem not only has clear real-world needs but is also an important research direction for ensuring reliable generalization of models in the real world. Based on this, we have supplemented the paper with an explanation of the AL+OOD application background to enhance the clarity and rationality of the task motivation.

---

> ### Author Response · Authors · 2025-12-04
> **W2 & Q1 [Data augmentation and AL can be mutually reinforcing when combined]**
>
> We thank the reviewers for their valuable questions. We believe that data augmentation and active learning are not two independent parallel techniques, but rather can form a dynamic positive synergy when used together. First, in the early stages when AL data is extremely scarce, the model's estimation of unlabeled samples is often not robust enough. Embedding augmentation into the AL process can stabilize the model's estimation before each round of sample selection, fundamentally avoiding the cumulative bias caused by early incorrect estimations in subsequent rounds. Second, the samples selected by AL usually have the highest information value. Applying augmentation to these key samples can amplify their contribution, generating samples with higher generalization gain, thereby further improving the selection quality of AL. Therefore, our method utilizes a dynamic coupling mechanism of "selection-augmentation-reselection," rather than a simple serial process. In our experiments, we systematically compared SimGAL with the "AL first, then augmentation" strategy. The results show that embedding augmentation into AL can significantly improve the effectiveness of the AL mechanism itself, outperforming the latter. Performing data augmentation after completing the AL process does lead to performance improvements, but the gains remain clearly inferior to those achieved when augmentation is integrated into the AL loop.
>
> | Method                      | (N_L=400)        | (N_L=600)        | (N_L=800)        | (N_L=1000)       |
> | --------------------------- | ---------------- | ---------------- | ---------------- | ---------------- |
> | AL first, then augmentation | 50.07 ± 2.15     | 55.38 ± 1.65     | 58.20 ± 0.95     | 59.41 ± 1.21     |
> | **SimGAL**                  | **53.56 ± 3.02** | **57.19 ± 2.37** | **60.74 ± 1.92** | **61.32 ± 1.35** |

---

> ### Author Response · Authors · 2025-12-04
> **W3 [Simplicity of a model does not equate to a lack of innovation.]**
>
> Thank you for the questions raised by the reviewers. We need to clarify that model simplicity does not equate to a lack of innovation. The core contribution of this paper lies not only in the specific design of the loss component, but also in the innovation of the problem perspective and the shift in the solution paradigm.
> 1 First, our core contribution is that we discovered the problem of insufficient generalization performance of the AL method, which poses a serious security risk when AL is applied to open-world (OOD) scenarios, a point often overlooked in previous research.
> 2 Second, our innovation is not merely the introduction of a loss component, but the construction of a closed-loop framework in which AL and data augmentation (DG) mutually promote each other. This design, which deeply embeds DG into the AL process, achieves a two-way gain of "augmentation correcting AL estimation bias" and "AL guiding high-value augmentation."
> 3 Finally, simplicity equals efficiency. We deliberately pursued a simple model design. Experiments demonstrate that this lightweight parameter learning component is sufficient to significantly improve the OOD generalization ability of AL without introducing complex computational burdens.

---

> ### Author Response · Authors · 2025-12-04
> **W4 [Semi-supervised methods are not good enough]**
>
> Thank you to the reviewers for their valuable questions. While using unlabeled data augments the training sample size, the distribution of this unlabeled data is identical to that of the labeled data, thus failing to effectively improve generalization ability for OOD. In section 4.3 of the paper, we report that even with the entire dataset labeled, our performance is inferior to SimGAL's performance with a very small number of labels. Furthermore, to ensure consistency with our settings, we compared the OOD performance of the semi-supervised method StyleMatch with our method. Experimental results show that using semi-supervised learning is inferior to our method.
> | Method       | NL=400          | NL=600          | NL=800          | NL=1000         |
> |--------------|------------------|------------------|------------------|------------------|
> | StyleMatch   | 42.18±1.42       | 45.72±1.74       | 47.61±2.32       | 49.26±0.94       |
> | SimGAL   | **53.56±3.02**   | **57.19±2.37**   | **60.74±1.92**   | **61.32±1.35**   |
>
>
> [1] Zhou K, Loy C C, Liu Z. Semi-supervised domain generalization with stochastic stylematch[J]. International Journal of Computer Vision, 2023, 131(9): 2377-2387.

---

> ### Author Response · Authors · 2025-12-04
> **Q2 [$N_L$ and $N_T$ of SimGAL represents different situations.]**
>
> Thank you to the reviewers for their valuable questions. N_T and N_L represent the alignment annotation amount and total sample size, respectively, indicating the comparison with the baseline under the same label budget and the same training sample size.

---

### Meta-Review · Area_Chair_BEAS · 2026-01-07

**Summary:**

The paper was reviewed by 4 experts and received ratings of 4424. The concerns of the reviewers are listed in the next field. Overall, the authors were able to address some concerns, while others still need further clarification and empirical support. Thus overall, the paper still would have had mixed ratings had the reviewers participated. The AC agrees with many of these concerns.

**Reviewer Concerns:**

**Reviewer LKR5 (rating 4)**
1. The DA approach to improve generalization is relatively independent from AL. Could do AL first, then the proposed method.
2. adding a loss-driven component on conventional DA is a bit simple and lacks innovation.
3. would OOD be better solved by using unlabeled data for unsupervised/semi-supervised learning?

The AC thinks the concerns were addressed reasonably well.

**Reviewer qY1K (rating 4)**
1. difficult to determine if improvement is due to DA or AL itself.  It is better to design new experiments that isolate the effects.
2. small-scale datasets - should try more challenging DG benchmarks like WILDS. The absolute OOD accuracy remains far from practical usability.
3. does pushing samples away from class centers potentially move them closer to other class centers?

The AC thinks that:
-  Point 1 was not addressed well; the question was about comparing fairly with augmentation-based AL baselines, e.g. LADA, using the same AL frameworks, so that the effect of the DA component can be isolated. In addition, there should be baseline experiments applying existing DG methods during AL. Table 3 seems close to this, but uses "random sample selection" rather than actual AL methods as in Table 2.
- Point 2 was not addressed well; Only 2 smaller datasets are presented.
- Point 3 was not addressed well; the response only states that the samples move away from the sample's class center and the distance is constrained, but this could inadvertently move the sample towards or even within the sphere of another class's center.

**Reviewer q11o (rating 4)**
1. Questionable setting - shouldn't adaptation from A to B be a better option?
2. uses pre-trained ResNet - is this setting meaningful?
3. not much theoretical analysis on the generalization setting.  lack of discussion about Bayesian AL.

The AC thinks that:
- Point 1 was not addressed particularly well. The situation raises additional questions about more suitable baselines, see Point 1 in qY1K discussion.
- Points 2 and 3 were addressed well.

**Reviewer w59N (rating 2)**
1. more empirical evidence comparing SGO to better-tuned DA pipelines.
2. how to optimize the augmentation parameters if the operations are not differentiable?
3. does not follow standard DG evaluation protocols.
4. missing related work.
5. does the ID performance also need to be maintained in the objective in Eq 2?
6. the connection between diversity and batch size is not clear.
7. how are class centers updated during training?

The AC thinks that:
- Point 1 was not addressed well by authors - in particular the tuning of hyperparameters of baselines.
- Point 3 was not addressed well -- source/target should be swapped and experiment run again to show generalizability.
- Points 2, 4, 5, 6, 7 were addressed well.

**Reviewer Scores:**

Reviewer LKR5 may have increased their score to marginal accept or higher.
Meanwhile, other reviewers still had outstanding issues that needed further clarification and experiments, and thus are unlikely to have raised their ratings.

---

### Decision · Program_Chairs · 2026-01-26

Reject